# Extracellular matrix and vascular dynamics in the kidney of a murine model for Marfan syndrome

**Rodrigo Barbosa de Souza**[1], **Renan Barbosa Lemes**[1], **Orestes Foresto-Neto**[2], **Luara Lucena Cassiano**[3], **Dieter P. Reinhardt**[4], **Keith M. Meek**[5], **Ivan Hong Jun Koh**[6], **Philip N. Lewis**[5]*, **Lygia V. Pereira**[1]*

1 Department of Genetics and Evolutionary Biology, University of São Paulo, São Paulo, SP, Brazil, 2 Faculty of Medicine, Department of Clinical Medicine, Renal Division, University of São Paulo, São Paulo, Brazil, 3 Department of Animal Sanity, Biologic Institute of São Paulo, São Paulo, SP, Brazil, 4 Department of Anatomy and Cell Biology Dentistry and Faculty of Dental Medicine and Oral Health Sciences, McGill University, Montreal, Quebec, Canada, 5 Structural Biophysics Research Group, School of Optometry and Vision Sciences, Cardiff University, Cardiff, United Kingdom, 6 Department of Surgery, Federal University of São Paulo, São Paulo, SP, Brazil

* lpereira@usp.br (LVP); lewispn@cardiff.ac.uk (PNL)

**Data Availability Statement:** All relevant data are within the manuscript and its Supporting Information files.

## Abstract

Fibrillin-1 is a pivotal structural component of the kidney's glomerulus and peritubular tissue. Mutations in the fibrillin-1 gene result in Marfan syndrome (MFS), an autosomal dominant disease of the connective tissue. Although the kidney is not considered a classically affected organ in MFS, several case reports describe glomerular disease in patients. Therefore, this study aimed to characterize the kidney in the mgΔ$^{lpn}$-mouse model of MFS. Affected animals presented a significant reduction of glomerulus, glomerulus-capillary, and urinary space, and a significant reduction of fibrillin-1 and fibronectin in the glomerulus. Transmission electron microscopy and 3D-ultrastructure analysis revealed decreased amounts of microfibrils which also appeared fragmented in the MFS mice. Increased collagen fibers types I and III, MMP-9, and α-actin were also observed in affected animals, suggesting a tissue-remodeling process in the kidney. Video microscopy analysis showed an increase of microvessel distribution coupled with reduction of blood-flow velocity, while ultrasound flow analysis revealed significantly lower blood flow in the kidney artery and vein of the MFS mice. The structural and hemodynamic changes of the kidney indicate the presence of kidney remodeling and vascular resistance in this MFS model. Both processes are associated with hypertension which is expected to worsen the cardiovascular phenotype in MFS.

## Introduction

The phenotype of Marfan syndrome (MFS) is defined by pleiotropic symptoms related to mutations in the *FBN1* gene, which encodes the fibrillin-1 protein in physiological settings [1]. The skeletal, ocular, and cardiovascular abnormalities have been referred to as the most common clinical findings in MFS [1,2]. Although alterations in other tissues/organs might be

**Funding:** This study was funded by Coordenação de Aperfeiçoamento de Pessoal de Nível Superior - Brazil (CAPES- Finance Code 001), Fundação de Amparo à Pesquisa do Estado de São Paulo (FAPESP), UK Medical Research Council (program grant MR/S037829/1) and the Canadian Institutes of Health Research (Grant PJT-162099). The funders had no role in study design, data collection and analysis, decision to publish, or preparation of the manuscript.

**Competing interests:** The authors have declared that no competing interests exist.

expected, based on this gene defect, these have been less exploited in the clinical setting [3]. Dysfunctions of the kidney, together with changes to the kidney's fibrillin-rich components have been reported in MFS patients as well as in several types of glomerular diseases [4–6], hydronephrosis [7,8], and renal cysts [3,9]. Other studies with mouse models of MFS showed hypoplasia glomerular in the hypomorphic model (mgR/mgR) [10] and cystic kidney disease in the C1039G mouse model [11].

The kidney plays a critical role in water and electrolyte balance. When the organ becomes dysfunctional, it can lead to hypertension through increased pressure on the arterial system [12–14]. Structurally, the kidney is a highly vascularized organ and receives 20% of the cardiac output [15]. The intricate vasculature is made up of arteries, arterioles, capillaries, and veins, many of which are associated with fibrillin-rich microfibrils and elastic fibers [16–19]. However, the variability of the kidney dysfunction phenotype in MFS patients makes it difficult to understand the etiology of the disease.

Mouse models of MFS provide a useful means of investigating the role of *Fbn1* mutations in kidney dysfunction. The mgΔ$^{lpn}$ is a variant of the mgΔ mouse model. In this model, 6 kb of the *Fbn1* gene, encompassing exons 19–24, were replaced with a neomycin resistance expression cassette [20] flanked by lox-P sequences [21]. The mgΔ$^{lpn}$ is a dominant-negative MFS model that exhibits a classic MFS-phenotype in heterozygosity with alterations in the skeletal, ocular, and cardiovascular systems [21–24]. In particular, mgΔ$^{lpn}$ mice develop aortic aneurysms and dissection similar to those seen in patients with a severe MFS phenotype [22].

Considering the close relationship between fibrillin-associated alterations in the vessel walls (that lead to disruption of morphology) and hemodynamics in MFS, the aim of this study is to characterize the extracellular matrix in the kidney as well as the macro and micro-hemodynamics of the renal vasculature in the mgΔ$^{lpn}$ MFS-model, in order to assess the possibility that the kidney is involved in the pathogenesis in MFS patients.

## Results

### Histology of the glomerulus

**The nephron is the functional filtration unit of the kidney and is composed of** the glomerulus and renal tubule [25]. The kidney is also divided into different regions, the renal cortex and the renal medulla [25,26] (Fig 1A). Histologically the renal cortex comprises the glomerulus, and the convoluted proximal and distal tubules [25]. In the MFS group the glomeruli exhibited variation in size. So, although some of the glomeruli were similar in size to the WT group, others showed a significant reduction in size (Fig 1B and 1C). To verify the integrity of the glomerulus structure in the MFS group (Fig 1C), we measured the area of the glomerulus (WT 1423μm$^2$ ± 655.6μm$^2$; MFS 681.7μm$^2$ ± 113,7μm$^2$) (Fig 1D), capillary space (WT 1142μm$^2$ ± 397.5μm$^2$; MFS 606μm$^2$ ± 108.9μm$^2$) (Fig 1E), and urinary space (WT 218μm$^2$ ± 250.7μm$^2$; MFS 138.7μm$^2$ ± 240.4μm$^2$) (Fig 1F). When compared to the WT group, the MFS group showed a significant reduction in all glomerular dimensions, including the glomerulus size.

### Fibers of the Elastic Fiber System (FEFS) in the glomerulus

Due to the significant size reduction of the glomerulus, and alterations in the FEFS in Marfan syndrome [27], we carried out a detailed ultrastructural analysis of the glomerulus by TEM to gain further insight to this structural alteration. For FEFS TEM analysis, tannic acid or orcein EM staining, which specifically enhances the FEFS components, was utilized [28–30].

The MFS group showed a reduced presence of FEFS and an increase in the presence of collagen fibers in the glomerular capsule (Fig 2B). A dysmorphology of the glomerular capillary region was also identified in the form of an amorphous substance between the mesangial area

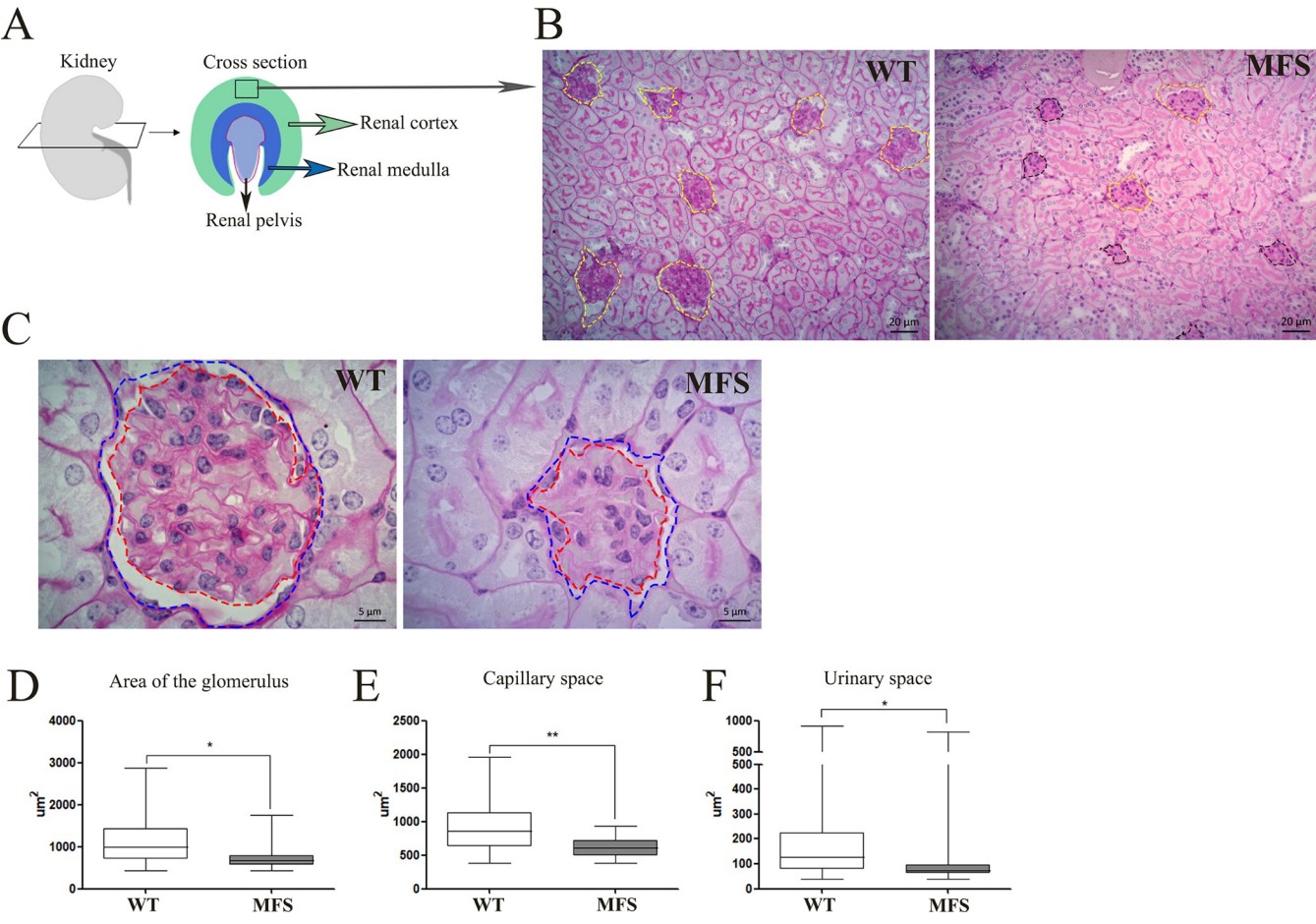

**Fig 1. Histology of glomerulus.** A. Scheme of the kidney region in cross-section. B. Histological analysis (PAS staining) at low magnification (B) and high magnification (C) of the renal cortex, yellow and black dotted lines delimit the normal glomerulus and reduced glomerulus respectively; dotted blue and red lines delimit the area of the glomerulus and of the glomerulus capillary, respectively. Area of the glomerulus (D); glomerulus-capillary (E); and urinary space (the space between the glomerulus and the glomerulus-capillary) (F) in WT mice (clear) and MFS mice (shaded). MFS mice showed a significant reduction in all areas measured when compared to WT. (*) p <0.05, (**) p<0,005. 10 WT and 10 MFS animals were used. The statistical analysis was performed by the Mann-Whitney test (B-D). Barr in B is 20μm and in C 5 μm.

and the glomerular capillary. Possibly, the amorphous substance inside the glomerulus near the glomerular capillaries is a basement membrane-like material (BM). This sort of substance has already been identified in glomerular basement membrane subendothelial alterations [4,31]. Further pathological changes in the capillary wall were also identified in the MFS group where duplication of the BM was evident in the kidney lesion described by Ghadially and Erlandson [32] (Fig 2C and 2D).

In the WT group, the FEFS were found to be localized close to the mesangial matrix and around the capillary walls. The FEFS exhibited a wide variability in diameter. In the MFS group, FEFS were also localized into the subendothelial regions close to the mesangial matrix, however, they exhibited a quite different FEFS morphology, comprising fragmented and thin fibers (Fig 2D).

Applying cluster analysis to determine the thickness of the FEFS in the WT group, we identified 3 separate subgroups: A, B, and C. Group A clustered 1nm-12nm (WT 5.87nm ± 3.51nm), group B 12nm-29nm (WT 20.77nm ± 4.11nm), and group C 29nm-280nm (WT 110.20nm ± 60.64nm). This separation would appear to indicate the presence of microfibrils (subgroup A), and elastic fibers (subgroup C) in the WT group as well as indicating the

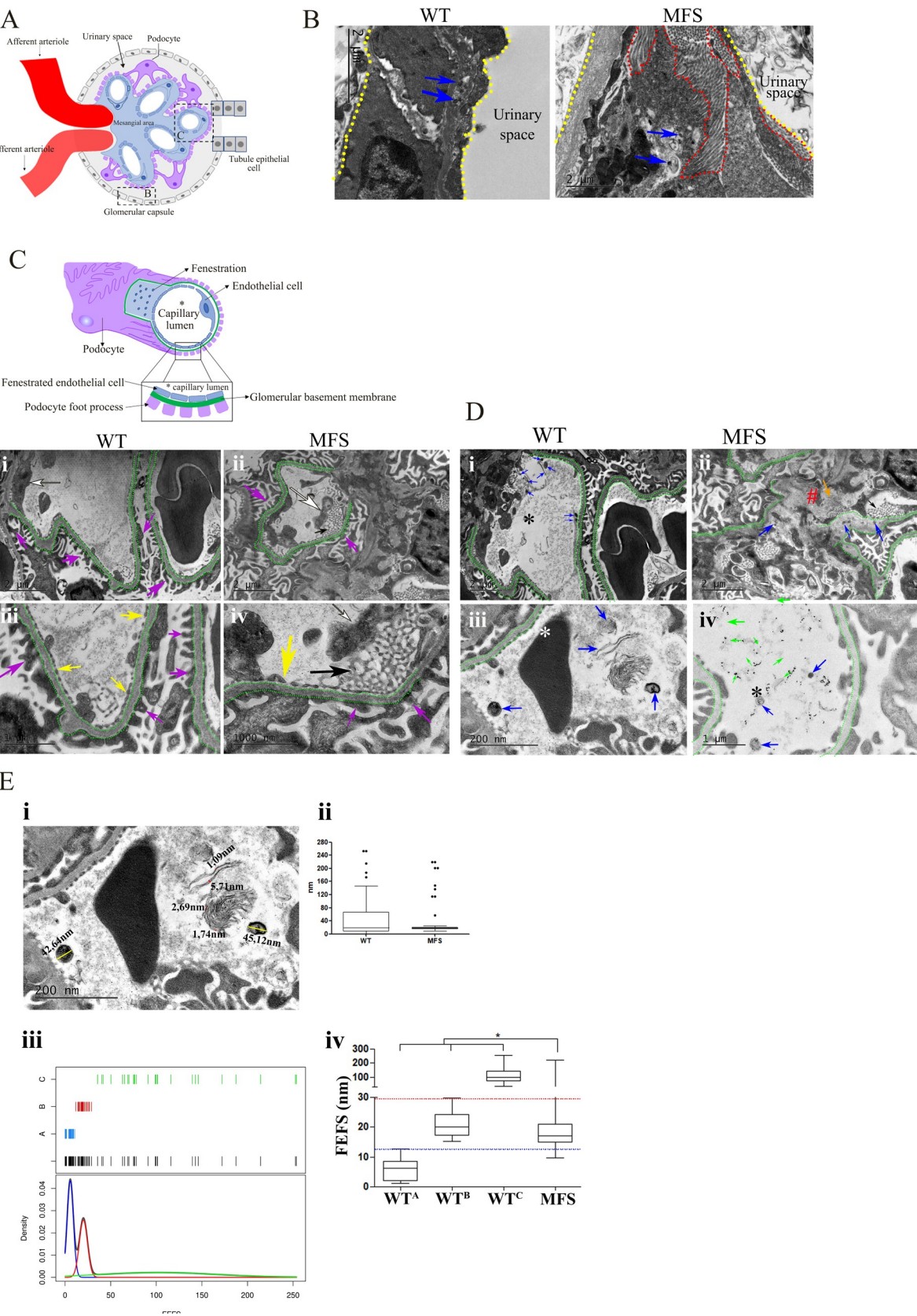

**Fig 2. Transmission electron microscopy of the glomerulus. A**. scheme of the glomerulus structures. **B**. TEM of the glomerular capsule with tannic acid staining enhancement of FEFS. The glomerular capsule is encircled by dotted yellow lines; the MFS group revealed an abundance of collagen fibers (encircled by dotted red lines). The FEFS are indicated with blue arrows. **C**. scheme of the glomerular capillary. i and iii are the WT group and ii and iv are the MFS group. The glomerular basement membrane of the capillary is marked with a dotted green line, the podocyte foot process with purple arrows, the fenestrated endothelial cell with yellow arrows, and the endothelial cell with a white arrow. The MFS group showed duplication of the basal lamina in the glomerular capillary (black arrow). **D**. i, ii, iii, and iv are TEM of the glomerular capillary; i and ii tannic acid staining, iii and iv orcein staining; the (*) represent the lumen of the glomerular capillary. FEFS (blue arrows) are localized beneath the endothelium of the glomerular capillary (green dotted line) and close to the mesangial matrix in the WT group. ii. In the MFS group, we observe FEFS (blue arrows) within a dense and amorphous component (orange arrow), together with an amorphous component in the space between capillaries (#) and duplication of the basal lamina in the glomerular capillary (black arrow); iii. The WT group showed FEFS (blue arrows). iv. In the MFS group, in addition to the integrity of FEFS (blue arrows), we observed many fragmented (green arrows) FEFS. **E**. FEFS analysis. i. WT TEM analysis with orcein technique exemplifying the measure of the FEFS; ii. FEFS thickness in both WT and MFS groups; iii cluster analysis in WT group, which separated the thickness into three sub-groups (WT[A], WT[B], and WT[C]); iv Analysis of the WT[A], WT[B], and WT[C] sub-groups and the MFS group showed a significant difference between MFS group when compared to WT sub-groups WT[A], WT[B], and WT[C]. (*) p <0.05. The statistical analysis was performed using cluster analysis as described in the Methods section. The dotted dark-blue and red lines represent the superior limits of the WT sub-groups A and B respectively. 3 WT and 4 MFS animals were used.

presence of complex fibers in the vascular pole inside the glomerulus. A significant difference between the 3 subgroups of the WT and the MFS group (MFS 30.51nm ± 44.95nm) was also identified (Fig 2E).

The MFS group showed a significant increase in FEFS thickness when compared to subgroup A of the WT. MFS group also exhibited a significant reduction of thickness when compared to subgroups B and C of the WT. These results suggest the MFS group did not have capacity to fully assemble the complex fibers of the FEFS (Fig 2E).

SBF-SEM was used to evaluate the 3D structural organization of the glomerulus and FEFS (Fig 3; S1 and S2 Videos). The WT group presented FEFS arranged in a tubular-shaped network within the capillary. In the MFS group FEFS were fractured and showed a loss of capillary structure (Fig 3). Changes in the capillary wall were also seen in the MFS group and appeared substantially closer to the intracapillary cells than in the WT group. This finding points to a loss of capillary elasticity, which could result in a decrease in hemodynamic flow.

Fibrillin-1 is a major component of the FEFS [27,33]. Immunofluorescence was therefore used to determine the fibrillin-1 distribution in the glomerular capillary. The MFS group showed a significant reduction in fibrillin-1 (WT 0.86 intensity/pixel ± 0.61 intensity/pixel; MFS 0.47 intensity/pixel ± 0.67 intensity/pixel) (Fig 4B and 4C), which suggests a loss of the pivotal FEFS fibrillar-structural in the glomerulus of the MFS mice. It has been suggested that fibronectin may be a critical prerequisite for fibrillin-1 assembly where it is considered to act as a template for microfibril deposition [27,34,35]. In this study, we observed a significant reduction of the fibronectin in the MFS group when compared to the WT group (WT 2.01 intensity/pixel ± 1.22 intensity/pixel; MFS 0.71 intensity/pixel ± 0.36 intensity/pixel) (Fig 4D and 4E); this finding suggests the reduction of the fibronectin could be associated with the loss of capacity to create complex FEFS in the glomerulus of MFS mice.

Both fibrillin-1 and fibronectin reduction in the MFS group could be linked to the proteolysis process. The MMP-9 is an enzyme, which shows elastolytic activity [36,37]. Immunofluorescence of MMP9, showed a significant increase in the MMP-9 intensity in the glomerulus and peritubular interstitium in the MFS group when compared to WT (WT 2.51 intensity/pixel ± 1.62 intensity/pixel; MFS 5.27 intensity/pixel ± 3.11 intensity/pixel) (Fig 4F and 4G). This suggests that active degradation of the FEFS in the MFS glomerulus is occurring.

## FEFS distribution in the peritubular interstitium

The components of the FEFS assembly like fibrillin-1, fibronectin, and MAGP (microfibrils associated with glycoprotein), were not only identified within the glomerulus but also in the

**WT**    **MFS**

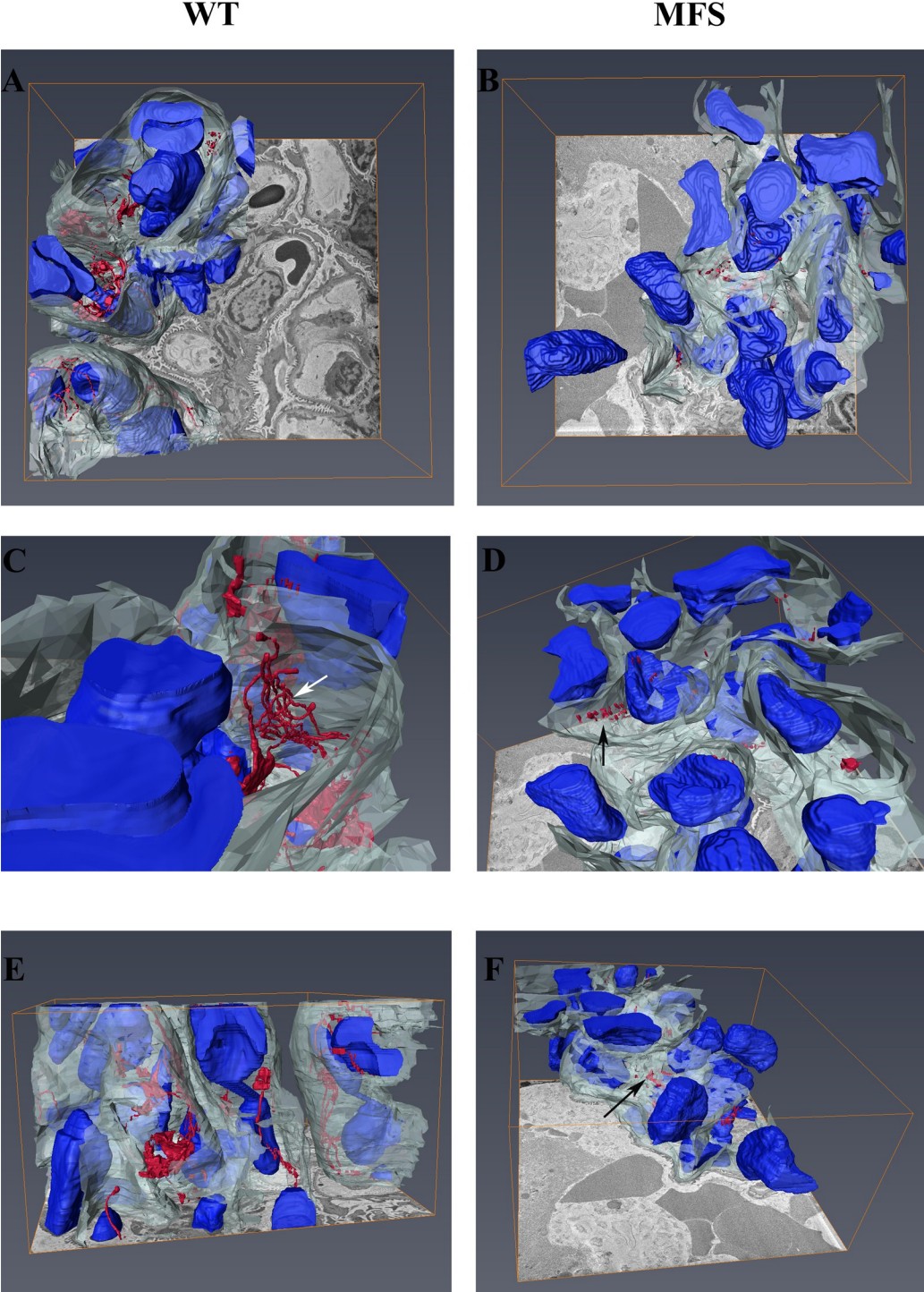

**Fig 3. 3D structural organization of the capillary of the glomerulus and microfibrils.** FEFS are shown in red; capillary in white and cells in blue. A, C, and E: WT group; B, D, and F: MFS group. A and B superior plane; C and D superior plane at higher magnification; E and F lateral plane. The WT group presents FEFS arranged in a tubular-shaped network (white arrow) within the capillary. The MFS group showed fractured FEFS (black arrow), and loss of capillary structure. 1 WT and 1 MFS animals were used.

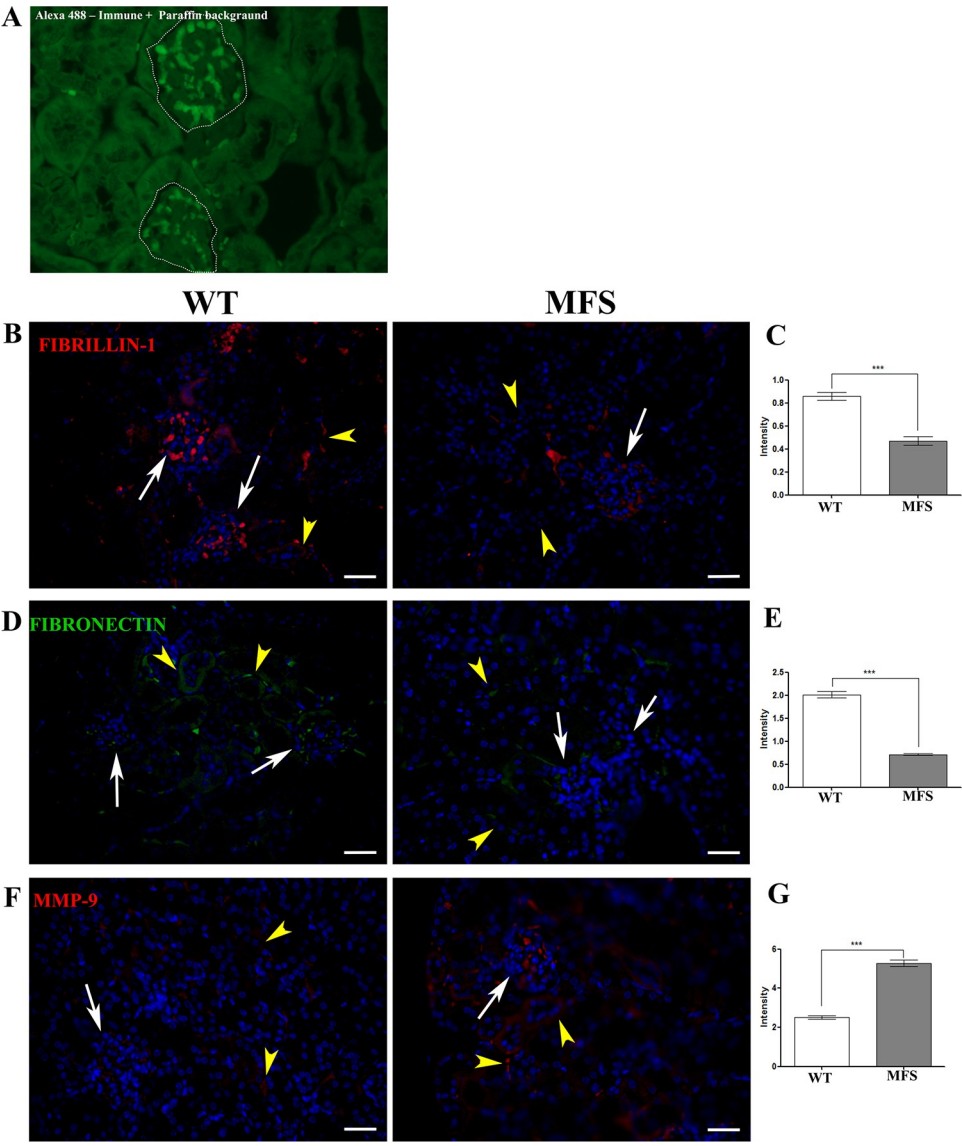

**Fig 4. Immunofluorescence for fibrillin-1, fibronectin, and MMP-9 in kidney tissue.** A. WT picture of excitation-emission for tissue visualization; the glomerulus is encircled by a dotted white line. In the WT group, the fibrillin-1 (B, red), and fibronectin (D, green) immunofluorescence displayed a wide distribution in the glomerulus (white arrows), and in the tissue between tubes (yellow arrowheads). The MFS group displayed only weak markers of the fibrillin-1 and fibronectin distributions, both within the glomerulus (white arrows) and in the tissue between tubes (yellow arrowheads). Quantification of the intensity of the fluorescence of the fibrillin-1 (C), and fibronectin (E) showed a significant reduction in the MFS group in both proteins. (F) MMP-9 in kidney tissue was weakly labeled in the WT group in the glomerulus (white arrow), and in the space between tubes (yellow arrowheads). The quantification of the fluorescence intensity of the MMP-9 (G), where the MFS group showed a significant increase when compared to the WT group. (***) p <0.001. 5 WT and 5 MFS animals were used. The statistical analysis was performed by the Mann-Whitney test (C, E, and G). Barr 5μm.

peritubular ECM [38]. It is worthy of note that TEM analysis revealed the presence of abundant FEFS in the WT group. In contrast, the MFS group revealed few and fragmented FEFS. Furthermore, the MFS group showed significant changes to the ECM, including a considerable increase in the presence of collagen fibers, indicative of a remodeling or fibrosis process in the kidney (Fig 5B–5D).

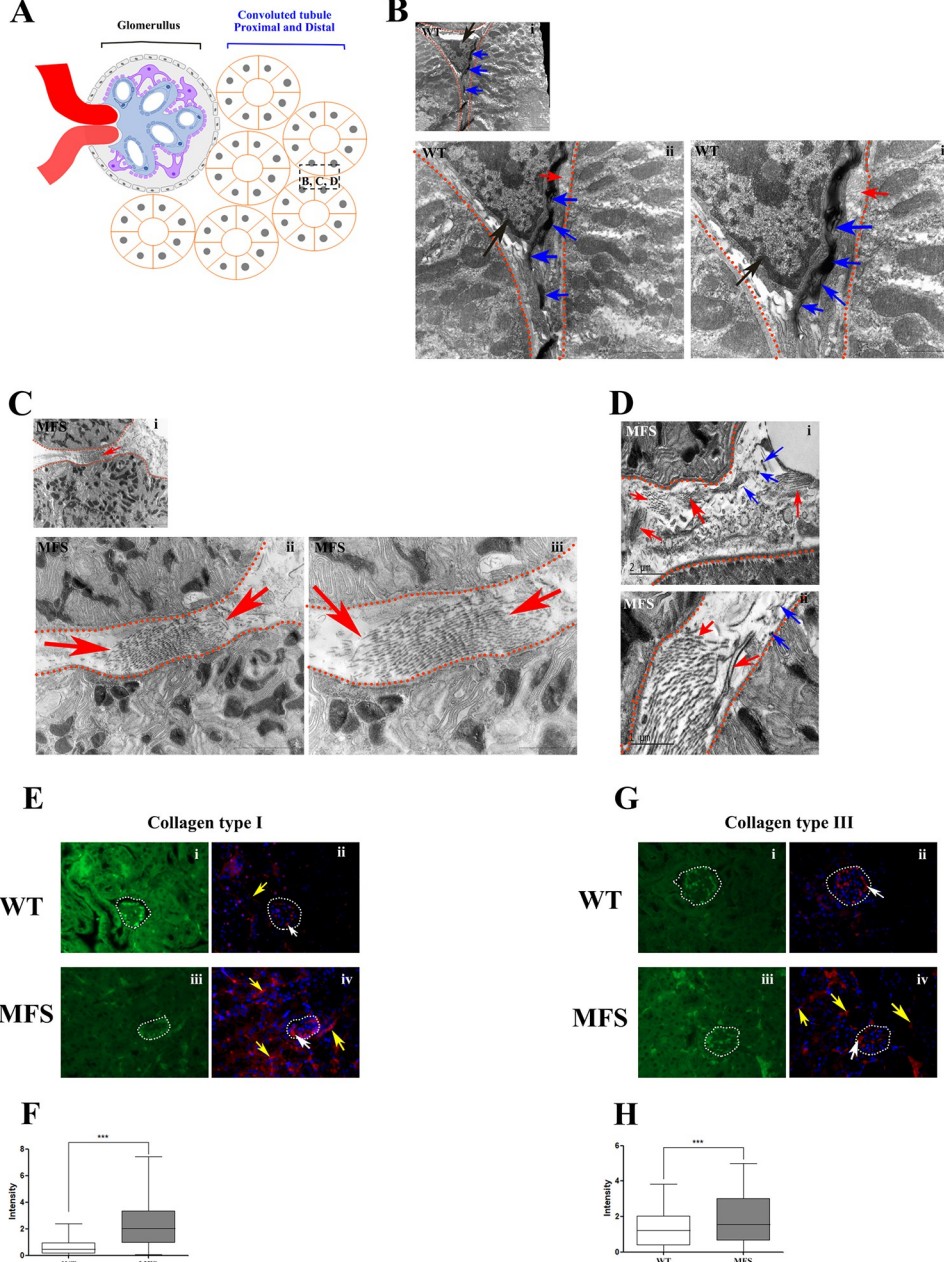

**Fig 5. TEM analysis of the FEFS and collagens fibers distribution in the peritubular interstitium. A**. Scheme of the glomerulus and convoluted tubule proximal and distal. **B**, **C**, and **D** Tannic acid staining enhancement of FEFS. The orange dotted lines delineate the epithelial tubes. **B**. WT group showed the presence of abundant FEFS (blue arrows), and few collagen fibers (red arrows) close to the cell (black arrow). In contrast, the MFS group (**C** and **D**) revealed only a few FEFS (blue arrows), nearly all of which were fragmented. Furthermore, the MFS group showed an increased presence of collagen fibers (red arrows). **E**. Immunofluorescence of collagen fibers type I in kidney tissue. The WT group (ii) showed few markers of the collagen type I (red) in the glomerulus (white arrow) and in the peritubular interstitium (yellow arrow); In the MFS group (iv) an abundance of collagen type I (red) labeling was observed in the peritubular interstitium (yellow arrows) and in the glomerulus (white arrow). **F**. Quantification of the fluorescence intensity of collagen type I, the MFS group exhibited a significant increase when compared to the WT group. **G**. Immunofluorescence of collagen fibers type III in kidney tissue. The WT group (ii) predominated within the glomerulus (white arrow), with few markers in the peritubular interstitium. The MFS group (iv) showed high levels of collagen type III (red) in the peritubular interstitium (yellow arrows), and in the glomerulus (white arrow). **H**. Quantification of the fluorescence intensity of collagen type III the MFS group showed a significant increase in intensity when compared to the WT group. The figures **E**-i and **E**-iii, and **G**-i and **G**-iii are in the Alexa 488 excitation-emission for visualization of the tissue. 3 WT and 4 MFS animals were used for TEM analysis; 5 WT and 5 MFS

animals were used for Immunofluorescence analysis. The statistical analysis was performed by the Mann-Whitney test (E and F). (***) p <0.001.

## Collagen distribution in the kidney

The kidney fibrosis process is a serious condition that develops as a result of an ECM imbalance and leads to organ malfunction and failure [39]. Collagen type I, type III, fibronectin, alpha-smooth muscle actin, and connexin 43 are among the proteins that arise throughout the process [40,41]. As a result, our research looked at the important proteins involved in the fibrosis process. Immunofluorescence of the MFS group revealed a significant increase in the presence of collagen types I (WT 0.61 intensity/pixel ± 0.54 intensity/pixel; MFS 2.34 intensity/pixel ± 1.64 intensity/pixel) and III (WT 1.28 intensity/pixel ± 0.93 intensity/pixel; MFS 1.80 intensity/pixel ± 1.30 intensity/pixel) in the glomerulus, peritubular interstitium, and periglomerular interstitium (Fig 5E, 5F, 5G and 5H) as compared to WT. WT showed the predominance of collagen type III mostly in the glomerulus (Fig 5G). TEM analysis confirmed these findings (Fig 2). The altered FEFS that we have demonstrated in the ECM of the MFS group, leads us to hypothesize that increased collagen fibers are deposited to support the structure of the ECM.

## A-smooth muscle actin, and connexin-43 in the kidney

α-smooth muscle actin levels have been reported to rise in response to injury or pathology, usually within myofibroblasts, which connect the cell with the ECM [40]. When compared to the WT group, the MFS group had a significant increase in α-smooth muscle actin (WT 12.61 intensity/pixel ± 4.71 intensity/pixel; MFS 16.92 intensity/pixel ± 5.11 intensity/pixel), which was widely disseminated in kidney tissue, suggesting that the MFS group may have damaged tissue (S1A, S1B and S1E Fig).

Cx43, whose expression increases in kidney fibrosis [41], was also studied. Cx43 was widely distributed in the glomerulus and in the cells of the parenchyma in both WT and MFS (S1C and S1D Fig). Interestingly, expression of Cx43 was found to be significantly reduced in the kidney tissue of the MFS group (WT 1.05 intensity/pixel ± 1.19 intensity/pixel; MFS 0.49 intensity/pixel ± 0.59 intensity/pixel) (S1F Fig).

## Kidney vessel integrity and blood flow

Following the examination of ECM components, we focused on vascular structural characteristics because of their inherent relationship to kidney ECM and vessel hemodynamics. The MFS group exhibited significantly reduced elastic fiber integrity (WT 0.89 ± 0.18; MFS 0.54 ± 0.28) and artery wall thickness in the tunica media (WT 15.98μm ± 2.34μm; MFS 14.28μm ± 1.78μm) (Fig 6A, 6B and 6C). Furthermore, blood flow through the renal artery was significantly reduced (WT 0.30 mL/min ± 0.05 mL/min; MFS 0.1 mL/min ± 0.03 mL/min) (Fig 6D) The renal artery is a direct branch of the abdominal aorta. We examined aorta blood flow, which was found to be significantly reduced in the MFS group as compared to the WT group (WT 3.18 mL/min ± 0.41 mL/min; MFS 2.07 mL/min ± 0.69 mL/min), in addition to having irregular peak systolic velocities and a lower systolic peak (S2 Fig). Based on these findings, we hypothesize that the lower blood flow in the renal artery is related to decreased aortic blood flow and loss of elastic fiber integrity.

We could not observe any quantitative differences in the wall structure of the renal vein in the MFS group (Fig 6E). In contrast, we detected a significant decrease in blood flow also in the renal vein (WT 0.31 mL/min ± 0.04 mL/min; MFS 0.07 mL/min ± 0.01 mL/min) (Fig 6F).

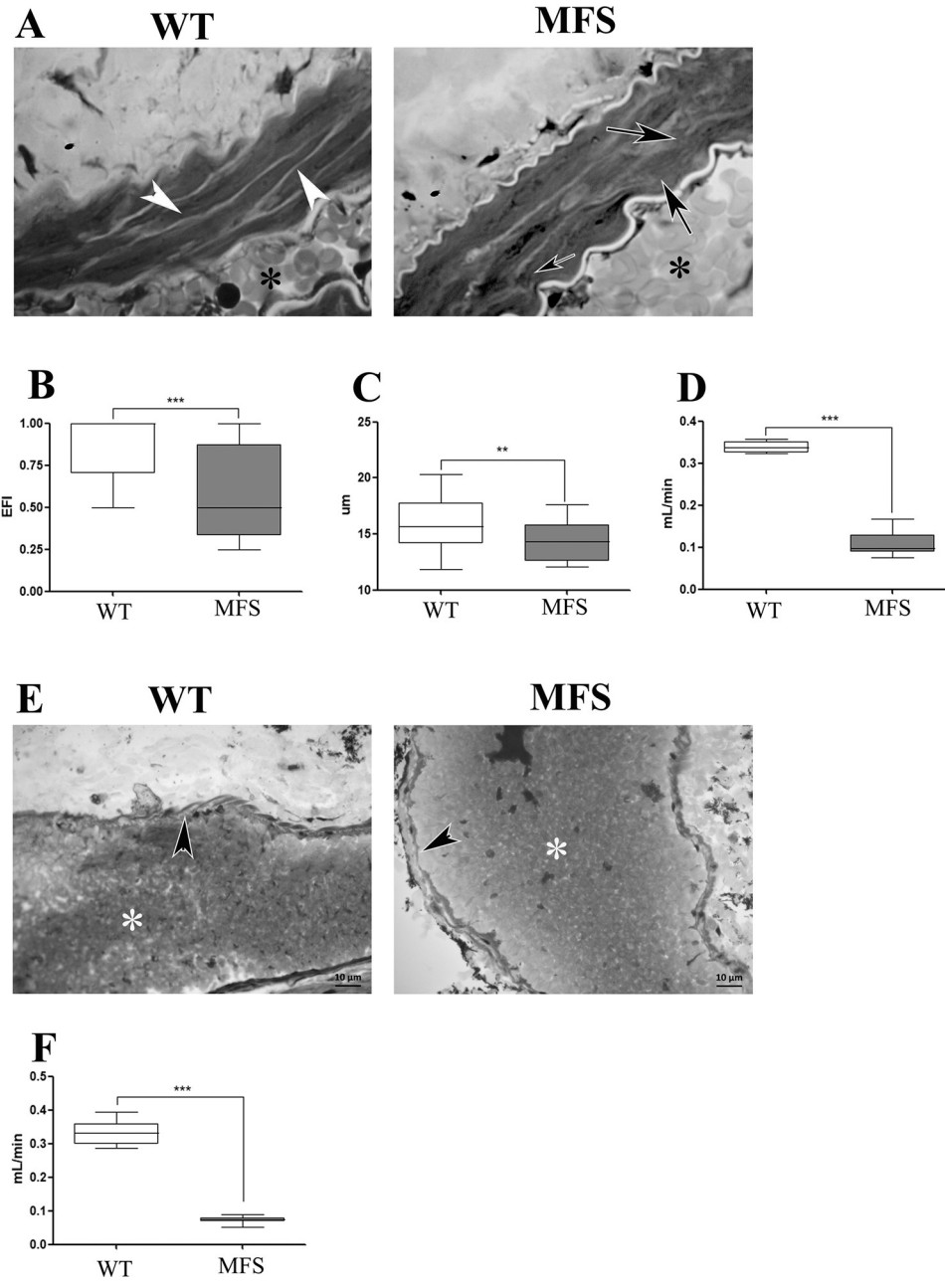

**Fig 6. Integrity of the renal artery and renal vein.** (A) Histological analysis of the renal artery in the WT group showed elastic fiber integrity around the smooth muscle cell (white arrowhead) in the tunica media. In the MFS group, fragmentation of the elastic fibers in the tunica media was observed (black arrow). B. EFI reveals a significant reduction in the MFS when compared to the WT group. C. Thickness of tunica media in the MFS group showed a significant reduction when compared to the WT group. D. Blood flow in the renal artery showed a significant reduction in MFS when compared to the WT group. E. Histological analysis of the renal vein. Tunica media is indicated by black arrowhead and lumen by an asterisk. F. Blood flow of the renal vein was significantly reduced in MFS when compared to the WT group. (**) p<0,005(***) p <0.001. 5 WT animals and 5 MFS animals were used for morphological study, and 10 WT animals and 10 MFS animals were used for hemodynamic study. The statistical analysis was performed by Mann-Whitney test. Barr 10μm.

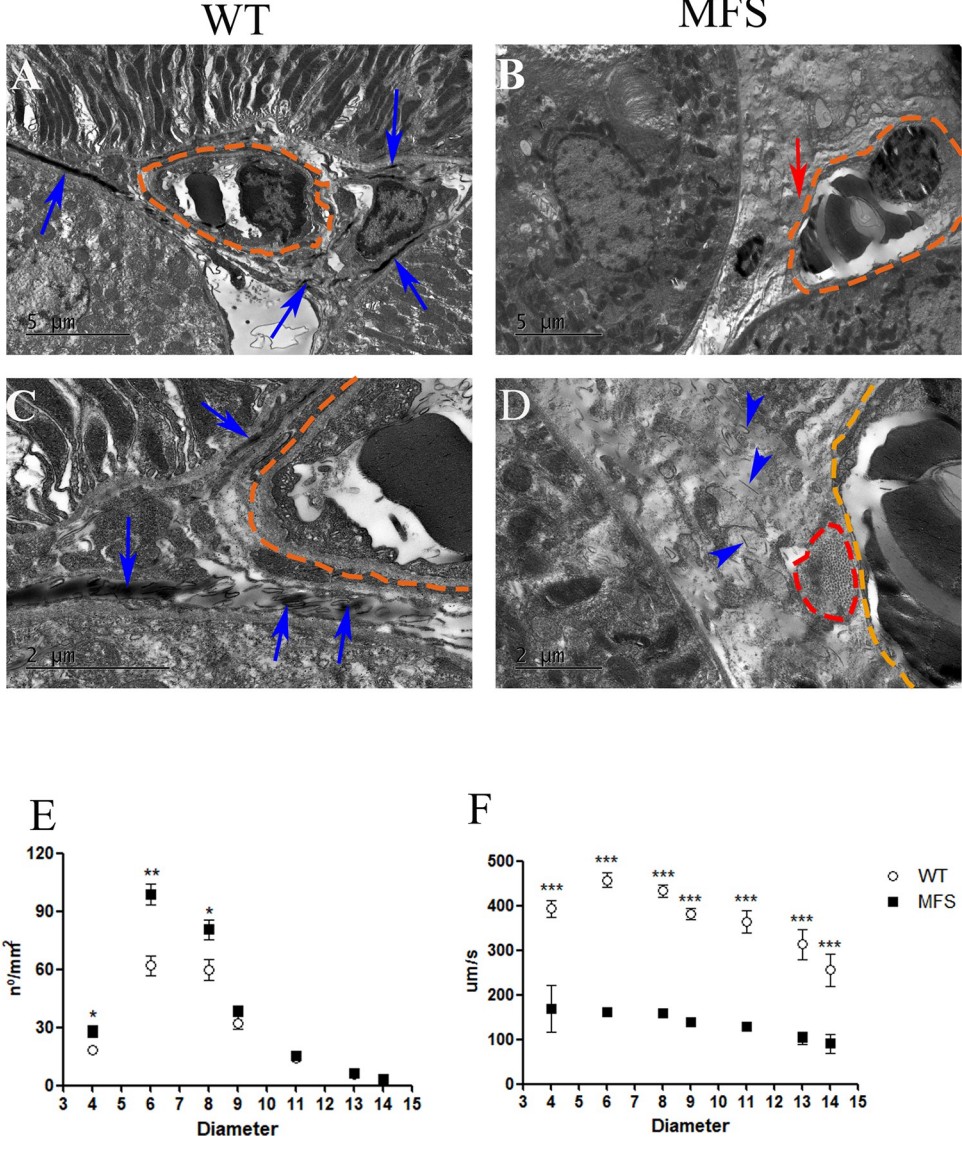

**Fig 7. Morphology and hemodynamics in the capillary.** A-D. Morphological analysis of the capillary. A and C: WT group, which showed densely packed FEFS (blue arrows) surrounding the capillary (orange dotted line). B and D: MFS group exhibited amorphous material around the capillary (orange dotted line), as well as FEFS fragmentation (blue arrowheads) and a cluster of collagens (red dotted line). E and F: Analysis of microhemodynamics. E Vascular density of the microvessel showed a significantly increased density in capillary vessels between 4μm to 8μm diameter in the MFS group when compared to the WT group. F. Velocity of the blood flow in microvessels between 4μm and 15μm diameter showed a significant reduction in the MFS group when compared to the WT group. (*) p <0.05, (**) p<0,005(***) p <0.001. 3 WT and 4 MFS animals were used for morphological study; 14 WT and 5 MFS animals were used for microhemodynamic study. Statistical analysis was performed by the Mann-Whitney test.

Finally, we analyzed the renal microvasculature. While in WT animals the pericapillary space presented FEFS integrity and absence of collagen (Fig 7A and 7C), MFS mice showed enlarged spaces between capillaries and tubules, with fragmented FEFS and focal clustering of collagen fibers (Fig 7B and 7D). These differences indicate compromised blood flow in MFS mice.

In vivo video microscopy of the kidney cortical microcirculation (S3 and S4 Videos) revealed a significant increase in the vascular density/diameter (Fig 7E) in the MFS group in

diameters 4μm (WT 14.37μm ± 5.17μm; MFS 28.2μm ± 6.61μm), 6μm (WT 62.91μm ± 20.02μm; MFS 98.60μm ± 11.95μm), and 8μm (WT 52.48μm ± 19.51μm; MFS 80.60μm ± 11.5μm) as compared to the WT. In contrast, the blood flow velocity decreased significantly in the MFS group in all diameters analyzed (WT 362.5 μm/s ± 128.5 μm/s; MFS 141.1 μm/s ± 78.6 μm/s) (Fig 7F). These findings reveal a substantial change in the microvessel dynamics in the MFS kidney.

## Kidney function and Angiotensin II

The MFS group showed alterations in renal ECM and hemodynamics. Therefore, we analyzed renal function by quantifying creatinine and urea in serum, the most commonly used biomarkers of kidney dysfunction. There was no significant difference in serum creatinine between the WT (0.60±0.12) and MFS groups (0.42±0.08). Similar findings were seen when serum urea was examined: the WT group had 61.09 mg/dL±11.54, whereas the MFS group had 54.66 mg/dL±8.77. When compared to a positive renal lesion control, both serum creatinine and serum urea exhibited lower creatinine levels (S3 Fig). The MFS group's renal function was preserved as a result of these findings.

Furthermore, due to significant reductions in blood flow in the MFS group, we analyzed angiotensin II (Ang II) in the kidney. Ang II is a major component of the renin-angiotensin-aldosterone system (RAAS), produced by convoluted tubules and juxtaglomerular cells [42,43]. Ang II in kidney tissue was predominantly identified in the convoluted tubules in WT. On the other hand, Ang II expression was significantly reduced in the MFS group (WT 166.6 intensity/pixel ± 10.68 intensity/pixel; MFS 152.7 intensity/pixel ± 12.41 intensity/pixel) (Fig 8).

## Discussion

Although kidney abnormality is not a typical phenotype in MFS patients, some patients exhibit kidney alterations [3–9]. The kidney is an important part of the vascular system as it is involved in the control of blood pressure. Therefore, the study of kidney in MFS may contribute to understanding the complete pathophysiology of the disease.

Mutation of the *Fbn1* gene in the mgΔ$^{lpn}$ model has previously been shown to cause changes in the fibrillin-1 network assembly [21]. The increased expression of MMP-9 in the MFS group in this study may also be implicated in the significant reduction of both fibrillin-1 and fibronectin, given that previous studies have shown in different connective tissue pathologies that MMP-9 plays a role in the degradation of fibrillin-1, fibronectin, and elastic fibers [36,44–47].

As fibronectin is a critical prerequisite for the fibrillin-1 network, microfibrils, and elastic fiber assembly [27,34,47,48], it would seem logical to consider that fibronectin reduction may directly impact the assembly of the fibrillin-1 and microfibrils. Atypical fiber phenotype formation was shown by both the ultrastructure 3d-reconstructions and the cluster analysis (an automated system without human input), where we observed in the MFS group fragmentation and significant variation of fiber thickness. These findings lead us to hypothesize that the instability in the formation of the complex fibers is directly related to the reduction of fibronectin and fibrillin-1.

Fibrillin-1 and fibronectin are both key structural components of the kidney [38,49,50]. Hence, the decrease in fibrillin-1 and fibronectin in the MFS mice may explain the glomerular hypoplasia. Interestingly, Hartner et al. [10] previously showed in the hypomorphic mgR/mgR MFS model that lower fibrillin-1expression, with a decrease in the immunolocalization of fibrillin-1 in the renal parenchyma, can lead to glomerular hypoplasia. However, even with

**A**

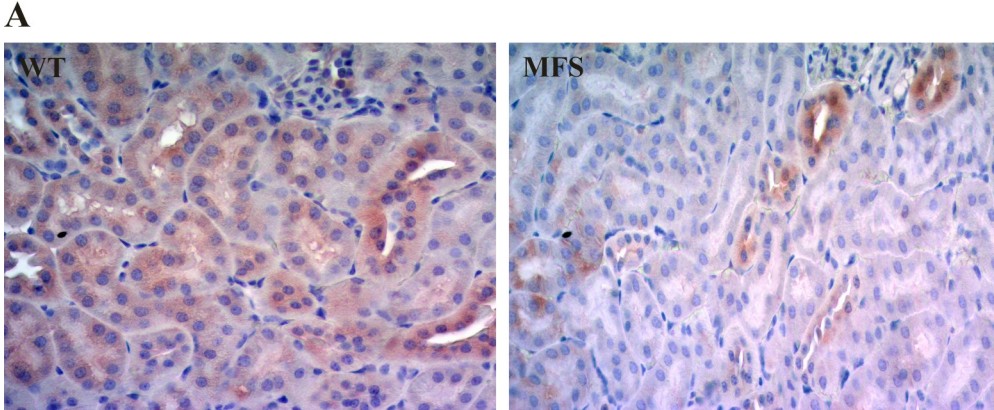

**B**

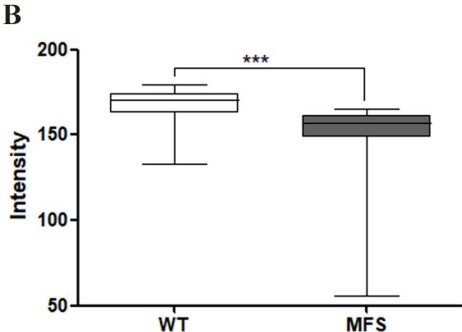

**Fig 8. Immunohistochemistry for Angiotensin II (Ang II) in kidney tissue.** A. Ang II (brown) in convoluted tubules in WT and MFS groups. There is strong wide distribution of Ang II in the WT group, while in the MFS group Ang II is present only in some convoluted tubules. B. Quantification of the intensity of the Immunohistochemistry of Ang II showed a significant reduction in the MFS group when compared to the WT group. (***) p <0.001. 3 WT and 4 MFS animals were used for the Immunohistochemistry study. The statistical analysis was performed by the Mann-Whitney test (C, E, and G).

these pathological changes to the glomerulus, no discernable impact on normal renal functional was evident. Our current findings would therefore support the hypothesis that alterations in fibrillin-1 are insufficient to cause a change in renal function, but can still cause an alteration in kidney structure.

An increase in collagen fibers and α-smooth muscle actin was also identified in the kidney of MFS animals, indicative of a fibrotic process [40]. Indeed, excessive collagen accumulation is known to occur in response to any type of tissue injury [51] and may lead to fibrosis [51–53]. Kidney fibrosis is the most common pathological manifestation of chronic kidney disease, which affect humans >65 years and known to be associated with organ malfunction and failure [39,54]. However, in this study, the animals were relatively young, which may explain the absence of critical levels of fibrosis and preserved kidney function.

Concomitantly, we investigated the expression of Cx43. Increased expression of this protein has been associated with chronic kidney disease and kidney fibrosis, and decreased expression with recovery of renal function and a reduced level of kidney fibrosis [41,55,56]. The MFS group showed a significant reduction of the Cx43 expression in kidney tissue, which suggests remediation of renal function despite the changes to the ECM.

The imbalance of the ECM in the MFS group suggests the participation of a compensating mechanism caused by the reduction/alteration in the FEFS. Nevertheless, it is likely that the accumulation of ECM in renal tissue could lead to changes in microvessel hemodynamics. The

MFS model exhibited duplication of the basal lamina in the glomerulus's capillary, suggestive of augmented stiffness of the capillary walls [32]. The MFS group also showed the accumulation of high-density amorphous material in the basement membrane of the glomerulus, similar to sclerotic lesions, a pathology previously described in MFS patients [4,31].

Since morphological alteration of microvessels was identified in the TEM analysis, we examined the microcirculation using SDF video-microscopy to determine any changes in flow dynamics. The SDF images provide dynamic images of the microcirculation, which show the 500μm deep penetration in the surface tissue with precise resolution exhibiting microvascular distribution and velocity [57]. The MFS group showed a significantly increased number of microvessels and a significant decrease in the velocity of blood flow.

We suggest that the increased collagen deposition in the renal ECM may limit microvessel compliance [40] leading to an increase in vascular resistance [58] reducing the flow velocity [59]. The increase in the density of small-diameter microvessels observed in the MFS group could be attributed to this occurrence, which could have resulted in the shunt mechanism reopening non-functioning microvessels [60].

The intense fragmentation of the elastic fiber in the renal artery of the MFS group might cause an alteration in the elasticity of the vessel and induce progressive tunica media stiffening and loss of the ability to distend during systole [58]. The fragmentation of elastic fibers in the aorta is usually related to an increase in collagen type I and enhanced vascular-wall stiffness in the MFS-murine model [22,61], and for that reason, we suggest this mechanism could also occur in the renal artery. The macro and microvessel alterations found in the MFS model suggest possible alterations in the microvessel and arterial aspects, which can be associated with cardiovascular alterations [62].

Additionally, in the MFS group, a significant reduction of angiotensin II (AngII) was found in the renal convoluted tubule, which at first raised the possibility of a direct connection with the blood flow and arterial tone alterations. Alterations in intrarenal AngII have been described in some physiological conditions, such as renal disfunction and hypertension. However, the precise contribution of intrarenal AngII in these conditions is poorly understood [63–66]. As we did not measure the plasmatic components of the Renin-Angiotensin-Aldosterone system (RAAS) and blood pressure, we could not conjecture their relationship with the hemodynamic state. In future studies, these pathophysiological aspects need further investigation.

Based on the overall findings, we propose that concomitant alterations in kidney vessels and tissues could be a contributing factor to understanding MFS phenotype and may have significant implications in the clinical management of the disease.

## Materials and methods

### Animals

Twenty-eight 6-month-old male mice of the C57Bl/6 strain were used. Of these, 14 animals were wild-type (WT) and 14 were heterozygotes from the mg$\Delta^{lpn}$ model (MFS) [21] of animals from different litters. The animals were maintained in collective cages at the Animal Facility of the Faculty of Medicine of the University of São Paulo, in an environment with an automated device that provides light/dark (6:00 am– 6:00 pm). The animals received a Labina® diet, water ad libitum. Euthanasia was performed after monitoring, by sectioning the abdominal aorta (near the branch of the right and left common iliac artery) and subsequent hypovolemic shock under general anesthesia (0.01/100mg Ketamine® and Xylazine® (4:1)).

This study was approved by the Institutional Animal Care and Use Committee of the Instituto de Biociências at the Universidade de São Paulo. Protocol ID: CEA/IBUSP 272/2016 Process 16.1.632.41.7.

## Morphological and morphometric study

Cross-sections of cortical kidney, renal artery, and renal vein samples were collected, fixed in 4% paraformaldehyde in 0.1 M PBS (pH 7.4), and embedded in resin (Technovit Kit 7100). Five slides of the cortical kidney from each animal were cut in series with four-micron-thick transverse slices and were stained with Toluidine Blue and Periodic Acid-Shiff (PAS). Four-micron-thick transverse slices were cut from the renal artery and renal vein and stained with Toluidine Blue. A Carl-Zeiss Axio Scope Microscope A1 was used to examine the samples.

The cortical kidney analysis was taken at 1000x magnification, and the area of the glomerulus, urinary space, and capillary space was measured using the "contours" tool of the ZEN software. The renal artery and renal vein were imaged at four different points at 400× magnification. Elastic fiber integrity was measured following the protocol used by Gyuricza et al. [67] and the thickness of the tunica media was measured using the "line" tool of the ZEN software.

## Electron microscopy ultrastructure analysis

Samples were prepared for serial block-face scanning electron microscopy (SBF-SEM) and transmission electron microscopy (TEM) using a staining method for elastic fibers developed by Lewis et al. [30]. Three 1mm$^3$ samples of the cortical kidney from each of three animals were fixed in modified Karnovsky's fixative (2.5% glutaraldehyde and 2% paraformaldehyde in 0.1M cacodylate buffer at pH 7.2) for 2 hours. After fixation, samples were washed in sodium cacodylate buffer 3 times for 10 minutes and distilled water for 5 minutes. The samples were post-fixed in 1% osmium tetroxide for 1 hour and washed with distilled water three times for 20 minutes. Subsequently, samples were put in 0.5% filtered tannic acid in distilled water for 2 hours, after which, samples were washed with distilled water three times over 30 minutes and left overnight in 2% aqueous uranyl acetate. Samples were then dehydrated in a 70–100% ethanol series. They were stained with 1% uranyl acetate for 2 hours followed by lead acetate in 1:1 ethanol and acetone and then placed into a 1:1 acetone and araldite resin mix (Araldite monomer CY212 and DDSA hardener) for 1 hour. BDMA accelerator was added to the pre-made Araldite resin making continuous resin changes to samples, 6 times for 2 hours each change. Samples were embedded in flat embedding molds and the blocks polymerized at 60˚C for 48 hours. Finally, the polymerized blocks were trimmed and mounted on Gatan 3view pins for either SBF-SEM or TEM.

To visualize elastic tissue, an en bloc Orcein staining method using TEM was performed [68]. Samples were fixed with 1% osmium tetroxide before being washed in distilled H$_2$O for 20 minutes and transferred to 70% ethanol for 10 minutes. Samples were stained with 0.3% orcein in 70% ethanol for 2 hours. After a 30 minutes wash with 70% ethanol, specimens were dehydrated in 90% ethanol for 20 minutes, followed by 100% ethanol × 2 for 20 minutes. Following dehydration, the tissue was subjected to the same remaining steps described in the tannic acid-based protocol.

The thickness of the fibers of the Elastic Fiber System (FEFS) were measured using the "line" tool of the ZEN software as describe by de Souza et al. [22].

## Serial Block Face Scanning Electron Microscopy (SBF-SEM)

Samples were examined using a Zeiss Sigma VP field emission gun scanning electron microscope (Carl Zeiss Meditec, Jena, Germany) equipped with a Gatan 3View2 system, where data sets of up to 1000 4K x 4K images were acquired every 75 nm at a pixel resolution of 7.3 nm. Three-dimensional (3D) reconstructions of data sets were created with Amira 6.0.1 software, using manual segmentation.

## Transmission electron microscopy

Ultrathin sections (90nm) were cut from the blocks mounted on 3view pins using a Leica UC6 ultramicrotome and were collected on copper grids (300 hexagonal mesh). They were imaged at an accelerating voltage of 80 kV using a JEOL 1010 transmission electron microscope fitted with a Gatan Orius 1000 camera (Gatan, Abingdon, England).

## Immunofluorescence

Left kidneys were cut coronally to identify the medullary and cortical regions. The cortical region was fixed overnight in 4% paraformaldehyde in PBS before paraffin embedding. Paraffin sections (6–8 μm thick) were collected on SuperFrost Plus slides (Thermo Fisher Scientific, Inc., Waltham, MA). Samples were dewaxed with xylols 3 times for 15 minutes at 56˚C and rehydrated through graded ethanol. Antigen retrieval was performed by heating the slides in citrate-EDTA buffer (10 mM citric acid, 2 mM EDTA, 0.05% Tween-20, pH 6.2) in a microwave oven three times for 1.5 minutes each at 50% power. Slides were washed twice for 2 minutes each in PBST (PBS containing 0.1% Tween-20) and for 5 minutes in PBS. Slides were incubated with blocking solution (10% normal bovine serum [NGS] in PBS) at room temperature for 1 hour and then with anti-fibrillin-1 [mrFbn1-C-74-F– 1:1000] antisera [69], anti-fibronectin [Abcam, ref. 45688–1:500], anti-MMP9 [Abcam, ref.38898 - 1:500], anti-collagen type I [Abcam, ref. 6308–1:500], anti-collagen typeIII [Abcam, ref. 7778–1:500], anti-alpha-smooth-muscle [Abcam, ref. 124964–1:500] and anti-conexin43 [Abcam, ref. 790101:500] antibodies diluted in blocking solution overnight at 4˚C. Sections were washed with PBST three times for 5 minutes each and incubated for 1 hour at room temperature with secondary antibody in blocking solution. Sections were washed three times for 10 minutes each in PBST and mounted in ProLong Gold Anti-Fading Reagent with 4′,6-diamidino-2-phenylindole (DAPI; Invitrogen), viewed and photographed using a Carl-Zeiss Imager.D2 microscope. The intensities of the color staining were analyzed with the "rectangle profile" tool of the Zen software.

## Immunochemistry

Left kidneys were cut coronally to identify the medullary and cortical regions. The cortical region was fixed overnight in 4% paraformaldehyde in PBS before paraffin embedding. Paraffin sections (6–8 μm thick) were collected on SuperFrost Plus slides (Thermo Fisher Scientific, Inc., Waltham, MA). Samples were dewaxed with xylols 3 times for 15 minutes at 56˚C and rehydrated through graded ethanol.

Antigen retrieval was performed by heating the slides in citrate buffer pH:6,0 in a steamer. The slides were washed once for 5 minutes each in PBS. Slides were incubated with blocking endogenous peroxidase (20mL hydrogen peroxide 30% in 180mL methanol) at a dark room for 30 minutes. After this, the slides were washed with once distilled water for 5 minutes. Then was washed with PBS for 5 minutes. The Immunochemistry was prepared as described [70]. Anti-Angiotensin II anti-body (Peninsula Lab. Int. antibody ref. #T4007) was used. The slides viewed and photographed using a Carl-Zeiss Axio Scope Microscope A1. The intensities of the color staining were analyzed with the "rectangle profile" tool of the Zen software was measured following the protocol used by Souza et al. [24].

## Macro- and micro-blood flow in the kidney

**Macro-blood flow.** Under general anesthesia (0.01/100mg Ketamine℞ and Xylazine℞ (4:1)). The abdominal aorta was dissected above the infra-phrenic artery near the aortic hiatus

and renal artery, after which an ultrasound flow-probe 2SB/T206 (Transonic Systems Inc, Ithaca, NY) was placed around the vessels. The microsurgery procedures were performed under a surgical microscope M900 D.F. Vasconcellos 1.2.6

**Micro-blood flow.**   The microcirculation of the kidney was evaluated using a Sidestream Dark-Field imaging device (MicroScanTM, MicroVision Medical Inc, Amsterdam, The Netherlands). The recordings were analyzed by the "enumerate space-time diagram" tool. In this study, the "Total Vessel Density", and the "Quantitative Velocity Assessment" were used for analysis. Vessel diameters were measured with a micrometer scale overlapped in the video display (S4 Fig).

## Serum creatinine and serum urea

The animals were anesthetized, and blood samples were taken. Serum creatinine was determined by Jaffé's modified method. Serum urea was measured using a commercially available kit (Labtest, Lagoa Santa, Brazil) in accordance with the manufacturer's instructions [71,72].

## Statistical analysis

The Shapiro-Wilk normality test was used to examine the results. A two-sample t-test was used if the data had a normal distribution. A Mann-Whitney U test was employed if the data did not have a normal distribution. The distribution of the FEFS was assessed by applying cluster analysis with the Mclust function, which divided the samples automatically into cluster samples. After this cluster analysis was applied, Kruskal-Wallis tests were used. When statistically significant p-values were produced, the post hoc Conover-Iman test of multiple comparisons [73] was applied and the corresponding p-values were adjusted using the Benjamini-Hochberg correction (False Discovery Rate—FDR) for multiple tests. All statistical analyses were performed in the R statistical package (Copyright (C) 2018 The R Foundation for Statistical Computing). All results were represented by mean and standard deviation. Differences were considered statistically significant at a $p$-value $< 0.05$.

## Limitations in research paper

The study was performed only in male mice. Considering the known differences in the cardiovascular system, and especially related to the RAAS this paper acknowledges this limitation.

## Supporting information

**S1 Checklist. The ARRIVE guidelines 2.0: Author checklist.**
(PDF)

**S1 Fig. Immunofluorescence for α-smooth muscle actin and connexin 43.** When compared to the WT group (A), the MFS group (B) showed increased staining for α-smooth muscle actin (red) in the glomerulus (white arrowhead), arteriole wall (yellow arrowhead), and myofibroblast cells (white arrows), Connexin-43 (purple) showed a wide distribution in the glomerulus (white arrowheads) and in cells in the parenchyma (white arrows) in the (C) WT group and in the (D) MFS group. Quantification of the intensity of the fluorescence of (E) α-smooth muscle actin and (F) connexin-43. The MFS group revealed a significant increase of α-smooth muscle actin and a significant reduction of Connexin-43. (\*\*\*) p <0.001. 5 WT and 5 MFS animals were used for Immunofluorescence analysis. The statistical analysis was performed by the Mann-Whitney test.
(TIF)

**S2 Fig. Aorta Blood Flow.** A. Spectral curve analysis of aortic blood flow, in the WT group, presents a uniform monophasic flow pattern with similar peak systolic velocities (PSVs) (black arrows) and no alterations in relaxation curves (black arrowhead). In MFS- we observe a PSV reduced (blue arrow), and alteration in the diastole curve (blue arrowhead). B. Quantification of the aorta blood flow revealed a significantly decreased in the MFS group when compared to the WT group. (*** $\rho<0.0001$) 10 WT and 10 MFS animals were used for hemodynamic study. The statistical analysis was performed by the Mann-Whitney test.
(TIF)

**S3 Fig.** A. serum creatinine of the WT, MFS groups, and positive control. We did not observe a significant difference between the WT and MFS groups. B. serum urea of the WT, MFS groups, and positive control. There was no significant difference between WT and MFS groups. 5 WT and 4 MFS animals were used for the kidney-function study. The statistical analysis was performed by the Mann-Whitney test (A and B).
(TIF)

**S4 Fig.** Steps of the processing SDF videos in Automated Vascular Analysis software (AVA) in both WT and MFS groups; i. video attached in AVA; ii. "Enumerate Space-time Diagram" tool in AVA, which identified microvessel; iii. After the "enumerate space-time diagram" tool the finalization of the analysis of the "Total Vessel Density" and "Quantitative Velocity Assessment.
(TIF)

**S1 Video. Illustrates the complex 3D arrangement of the glomerulus and microfibers in the WT.**
(MP4)

**S2 Video. Illustrates the complex 3D arrangement of the glomerulus and microfibers in the MFS.**
(MP4)

**S3 Video.** *In vivo* **videomicroscopy of the kidney cortical microcirculation in the WT.**
(AVI)

**S4 Video.** *In vivo* **videomicroscopy of the kidney cortical microcirculation in the MFS.**
(AVI)

**S1 File.**
(PDF)

## Acknowledgments

In addition, we thank the Central de Aquisição de Imagens e Microscopia (CAIMi-IBUSP), São Paulo, Brazil for acquisition of the transmission electron microscopy pictures.

## Author Contributions

**Formal analysis:** Rodrigo Barbosa de Souza, Renan Barbosa Lemes, Orestes Foresto-Neto, Luara Lucena Cassiano, Keith M. Meek, Ivan Hong Jun Koh, Philip N. Lewis.

**Funding acquisition:** Keith M. Meek, Lygia V. Pereira.

**Investigation:** Rodrigo Barbosa de Souza, Philip N. Lewis.

**Methodology:** Rodrigo Barbosa de Souza, Renan Barbosa Lemes, Orestes Foresto-Neto, Luara Lucena Cassiano, Philip N. Lewis.

**Resources:** Dieter P. Reinhardt, Keith M. Meek, Ivan Hong Jun Koh, Lygia V. Pereira.

**Supervision:** Keith M. Meek, Ivan Hong Jun Koh, Lygia V. Pereira.

**Validation:** Philip N. Lewis.

**Visualization:** Rodrigo Barbosa de Souza, Philip N. Lewis.

**Writing – original draft:** Rodrigo Barbosa de Souza, Dieter P. Reinhardt.

**Writing – review & editing:** Renan Barbosa Lemes, Keith M. Meek, Ivan Hong Jun Koh, Philip N. Lewis, Lygia V. Pereira.

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
