## [Decision Letter · Decision Letter 0]

21 Nov 2022

PONE-D-22-28441Extracellular Matrix and Vascular Dynamics in the Kidney of a murine model for Marfan syndromePLOS ONE

Dear Dr. Pereira,

Thank you for submitting your manuscript to PLOS ONE. After careful consideration, we feel that it has merit but does not fully meet PLOS ONE’s publication criteria as it currently stands. Therefore, we invite you to submit a revised version of the manuscript that addresses the points raised during the review process.

We look forward to receiving your revised manuscript.

Kind regards,

Peter R. Corridon

Academic Editor

PLOS ONE

Journal Requirements:

2. As part of your revision, please complete and submit a copy of the Full ARRIVE 2.0 Guidelines checklist, a document that aims to improve experimental reporting and reproducibility of animal studies for purposes of post-publication data analysis and reproducibility: https://arriveguidelines.org/sites/arrive/files/documents/Author%20Checklist%20-%20Full.pdf (PDF). Please include your completed checklist as a Supporting Information file. Note that if your paper is accepted for publication, this checklist will be published as part of your article.

"We are grateful to Coordenação de Aperfeiçoamento de Pessoal de Nível Superior - Brazil

(CAPES) – Finance Code 001; Fundação de Amparo à Pesquisa do Estado de São Paulo

(FAPESP), and UK Medical Research Council (programme grant MR/S037829/1) that financed

this study. Futhermore, we appreciate the Canadian Institutes of Health Research (Grant PJT162099) that enabled the generation of the anti-fibrillin-1 antiserum.

In addition, we thank the Central de Aquisição de Imagens e Microscopia (CAIMiIBUSP), São Paulo, Brazil for acquisition of the transmission electron microscopy pictures"

"Coordenação de Aperfeiçoamento de Pessoal de Nível Superior - Brazil (CAPES) – Finance Code 001.

Fundação de Amparo à Pesquisa do Estado de São Paulo (FAPESP)

UK Medical Research Council (programme grant MR/S037829/1) that financed this study. 

Canadian Institutes of Health Research (Grant PJT-162099) that enabled the generation of the anti-fibrillin-1 antiserum."

Reviewers' comments:

Reviewer's Responses to Questions

**Comments to the Author**

1. Is the manuscript technically sound, and do the data support the conclusions?

Reviewer #1: Yes

Reviewer #2: Partly

2. Has the statistical analysis been performed appropriately and rigorously? 

Reviewer #1: I Don't Know

Reviewer #2: Yes

3. Have the authors made all data underlying the findings in their manuscript fully available?

Reviewer #1: Yes

Reviewer #2: Yes

4. Is the manuscript presented in an intelligible fashion and written in standard English?

Reviewer #1: Yes

Reviewer #2: Yes

5. Review Comments to the Author

Reviewer #1: The aim of this study is the characterization of the renal phenotype in a relatively new Marfan mouse model (delta lpn) to assess whether the kidney is directly or indirectly is involved or not in the cardiovascular pathogenesis in MFS patients.

The article is of interest for readers involved in the field as well as it is well performed and relatively well explained. In this respect, the Introduction requires more work explaining more the model and other studies involving renal dysfunctions in other mouse models (i.e. PMID: 30359839) and it is not necessary that explains in the introduction the main obtained findings.

There is two missing points that should be addressed in some extent in the manuscript, which is mostly simply descriptive (which is fine). Is there some inflammatory response in affected glomeruli? How is the blood pressure in these Marfan mice? Are the angiotensin II blood levels altered? It would be greatly interesting to include some of this data in the paper.

One point that is not well explained is when authors affirm that ftagmentation of elastic fibers in the aorta is related to an increase in collagen type I…(Discussion section). In fact, usually (not always) collagen expression is increased to compensate in some extent the loss of elasticity.

Please, next time, put the page number.

Reviewer #2: This is an interesting study providing compelling evidence of alterations in kidney morphology in a mutant fibrillin-1 mouse model. The authors clearly show that glomerular morphology is affected, and that fibrillin and elastin fibers are damaged in the Fbn1 mutants, associated with an increased deposit of collagen fibers. Renal blood flow was significantly reduced in the Fbn1 mutants, associated with an increase in microvascular density. Nevertheless, kidney function was apparently not affected in the mutant mouse model.

While the morphological phenotype in the kidney of the Fbn1 mutant mice is clearly demonstrated, it is surprising that no functional differences were shown. The filtration function of the glomeruli does not appear to be decreased in the Fbn1 mutant mice, since serum creatinine and urea are not different. The authors do not clearly address the possible reasons why this discrepancy exists between the striking morphological phenotype together with the strong reduction in renal blood flow and the lack of a functional phenotype in the kidney. Besides the important filtration function, the kidney also plays a central role in blood pressure regulation. Renal blood flow is known to affect renin production. Did the authors look at activity of the RAAS in the Fbn1 mutant mice? At a minimum, blood pressure measurements should be performed to investigate whether the reduction in renal blood flow leads to angiotensin-induced hypertension. This could be an important mechanism which might contribute to the cardiovascular pathophysiology in patients with Marfan syndrome (e.g. as described in DOI: 10.1007/s004670050327). The authors make a similar statement at the end of the discussion, but this is currently not supported by the data presented in the manuscript.

A minor comment relates to the choice of the mgΔlpn Fbn1 mutant model for this study. What was the rationale of the authors to use this specific model instead of more commonly used Fbn1 mutant mouse models? It would also be interesting to provide a more direct comparison to the previously published findings in the mgR/mgR model (DOI: 10.1007/s00428-004-1081-6), which seems quite similar to the phenotype described in this study.

Other minor comments:

- In the results section on “Fibers of the Elastic Fiber System (FEFS) in the glomerulus”, the authors mention the existence of “an amorphous substance between the mesangial area and glomerular capillary”. Can they elaborate on the potential type of substance observed here?

- In the cluster analysis of the FEFS, the authors state that subgroup A consists of microfibrils, and subgroup C of elastic fibers. What does subgroup B represent? And why were the MFS FEFS not divided in corresponding subgroups for the comparisons in Figure 2?

- From the histological analysis shown in panel A of Figure 6, the fragmentation of the elastic fibers is not very clear. Perhaps a more detailed view can be shown, or the elastic fibers can be marked more clearly?

- A limitation of the study is that only male mice were examined. Considering the known differences in the cardiovascular system, and especially related to the RAAS the authors should at least acknowledge this limitation.

6. PLOS authors have the option to publish the peer review history of their article (what does this mean?). If published, this will include your full peer review and any attached files.

Reviewer #1: No

Reviewer #2: No

---

## [Author Response · Author response to Decision Letter 0]

4 Jan 2023

Extracellular Matrix and Vascular Dynamics in the Kidney of a murine model for Marfan syndrome

Rodrigo Barbosa de Souza1; Renan Barbosa Lemes1; Orestes Foresto-Neto3; Luara Lucena Cassiano4; Dieter P Reinhardt 5; Keith M. Meek2; Ivan Hong Jun Koh6; Philip N. Lewis*2; Lygia V Pereira1.

5. Review Comments to the Author

We appreciate the Reviewers' efforts in providing a detailed analysis of our paper and thank them for their constructive ideas for its development. To accommodate all of their recommendations, we made significant adjustments to the text, which we discuss in full below. We hope that our paper is now of sufficient high quality to be published as a PLoSOne.

Reviewer #1: The aim of this study is the characterization of the renal phenotype in a relatively new Marfan mouse model (delta lpn) to assess whether the kidney is directly or indirectly is involved or not in the cardiovascular pathogenesis in MFS patients.

The article is of interest for readers involved in the field as well as it is well performed and relatively well explained. In this respect, the Introduction requires more work explaining more the model and other studies involving renal dysfunctions in other mouse models (i.e. PMID: 30359839) and it is not necessary that explains in the introduction the main obtained findings.

There is two missing points that should be addressed in some extent in the manuscript, which is mostly simply descriptive (which is fine). Is there some inflammatory response in affected glomeruli? How is the blood pressure in these Marfan mice? Are the angiotensin II blood levels altered? It would be greatly interesting to include some of this data in the paper.

One point that is not well explained is when authors affirm that fragmentation of elastic fibers in the aorta is related to an increase in collagen type I…(Discussion section). In fact, usually (not always) collagen expression is increased to compensate in some extent the loss of elasticity

Please, next time, put the page number.

We are most gratified by the Reviewer’s view about our paper “The article is of interest for readers involved in the field as well as it is well performed and relatively well explained”.

The Reviewer makes 3 comments relating to adding new text in the introduction. 

We appreciate the feedback and have added more details about our model (lines 130–136). Next, in lines 116–122, we comment on the renal impairment that has been seen in other experimental models. In addition, the introduction's data findings were removed.

Lines 130 to 136. Mouse models of MFS provide a useful means of investigating the role of Fbn1 mutations in kidney dysfunction. The mg∆lpn is a variant of the mg∆ mouse model. In this model, 6 kb of the fibrillin-1 gene, encompassing exons 19-24, were replaced with a neomycin resistance (neoR) expression cassette (Pereira et al., 1997) flanked by lox-P sequences (Lima et al., 2010). The mg∆lpn is a dominant-negative MFS model that exhibits classic MFS-phenotypes in heterozygosity with alterations in the skeletal, ocular, and cardiovascular systems (Lima et al., 2010; de Souza et al., 2019; Souza et al., 2021a; Souza et al., 2021b). In particular, mg∆lpn mice develop aortic aneurysms and dissection, similar to those seen in patients with a severe MFS phenotype (de Souza et al., 2019). 

Lines 116 to 122. … Although alterations in other tissues/organs might be expected based on this gene defect, these have been less exploited in the clinical setting [3]. Dysfunctions of the kidney, together with changes to the kidney's fibrillin-rich components by mutations to the FBN1 gene, have been reported in MFS as well as in several types of glomerular diseases [4,5,6], hydronephrosis [7,8], and renal cysts [3,9]. Other studies with MFS mouse models showed hypoplasia glomerular in a hypomorphic mouse model (mgR/mgR) and cystic kidney disease in the C1039G mouse model (Hartner et al., 2004; Hibender et al., 2019).

In addition, the Reviewer makes 3 questions about the paper.

Is there some inflammatory response in affected glomeruli? 

In our study, we found no polymorphonuclear or mononuclear inflammatory infiltrates in the renal parenchyma.

The fibrillin-1 mutation has been linked to Marfan syndrome and has the ability to affect TGF-β signaling, promoting its activation (Neptune et al., 2004; Milewicz et al., 2021). TGF-β is a cytokine that, in addition to stimulating macrophages, has been linked to an anti-inflammatory mechanism (Fadok et al., 1998; Arabpour et al., 2021). As a result, we propose that the absence of polymorphonuclear or mononuclear inflammatory infiltrates could be associated with TGF-β activation in MFS. (Milewicz et al., 2021).

Neptune ER, Frischmeyer PA, Arking DE, Myers L, Bunton TE, Gayraud B, Ramirez F, Sakai LY, Dietz HC. Dysregulation of TGF-beta activation contributes to pathogenesis in Marfan syndrome. Nat Genet. 2003 Mar;33(3):407-11. doi: 10.1038/ng1116.

Milewicz DM, Braverman AC, De Backer J, Morris SA, Boileau C, Maumenee IH, Jondeau G, Evangelista A, Pyeritz RE. Marfan syndrome. Nat Rev Dis Primers. 2021 Sep 2;7(1):64. doi: 10.1038/s41572-021-00298-7.

Fadok VA, Bratton DL, Konowal A, Freed PW, Westcott JY, Henson PM. Macrophages that have ingested apoptotic cells in vitro inhibit proinflammatory cytokine production through autocrine/paracrine mechanisms involving TGF-β, PGE2, and PAF, J. Clin. Invest., 1998, vol. 101 (pg. 890-898).

Arabpour M, Saghazadeh A, Rezaei N. Anti-inflammatory and M2 macrophage polarization-promoting effect of mesenchymal stem cell-derived exosomes. Int Immunopharmacol. 2021 Aug;97:107823. doi: 10.1016/j.intimp.2021.107823.

How is the blood pressure in these Marfan mice? 

Despite the strong correlation between renal injury and blood pressure, mean arterial pressure was not evaluated in this study due to the possibility that invasive probe placement (Drost CJ 2019) could negatively affect renal micro- and macrocirculatory hemodynamics. However, the blood flow and spectral curve of the abdominal aorta were evaluated by placing a periaortic probe around it right above the inferior infra-phrenic artery. 

Data analysis showed a significant reduction in aortic flow in the MFS group compared to the wild-type group (Figure below - B), thus justifying the reduction in renal artery flow in the MFS group. We have previously reported the aorta blood flow reduction in mg∆lpn female mice (de Souza et al 2019) and it was attributed to high-output cardiac failure.

Additionally, we noticed continuous peak systolic velocity in the WT group as well as constant monophasic flow patterns and no alterations in the patterns of the relaxation curves. As opposed to the WT group, the MFS group displayed monophasic flow with variable peak systolic velocities and a lower systolic peak. Moreover, it was followed by altered diastolic phase peaks (Figure below - A).

Drost CJ. This handbook is an educational service provided. Transonic Systems Inc.®.2019.

Supplementary figure 2. Aorta Blood Flow. A. Spectral curve analysis of aortic blood flow, in the WT group, presents a uniform monophasic flow pattern with similar peak systolic velocities (PSVs) (black arrows) and no alterations in relaxation curves (black arrowhead). In MFS- we observe a PSV reduced (blue arrow), and alteration in the diastole curve (blue arrowhead). B. Quantification of the aorta blood flow revealed a significantly decreased in the MFS group when compared to the WT group. (*** ρ<0.0001) The statistical analysis was performed by the Mann-Whitney test

Are the angiotensin II blood levels altered? It would be greatly interesting to include some of this data in the paper

We are grateful for your suggestion and comments on adding the angiotensin II (Ang II) measure for improvement in our study.

Animals' plasma was not stored, making it impossible to measure the plasma levels of angiotensin II suggested by the referee. Therefore, we used the Zen software to determine the intensity of angiotensin II expression in kidney tissue via immunohistochemistry.

The marker of Ang II in kidney tissue was predominantly identified in the convoluted tubules in WT. On the other hand, Ang II expression was significantly reduced in the MFS group as compared to the WT.

These findings were consistent with those of Fountain and Lappin (2022), who observed a reduction of Ang II by angiotensinogen-converting enzyme inhibitors and angiotensin receptors, leading to reduction in blood pressure, in blood flow, and in arterial tones. Based on these findings, the significant reduction of Ang II in the MFS group appears to explain the reduction of renal artery blood flow and velocity in microcirculation in mgΔlpn mice, as well as reductions of the systolic peak in the aortic spectral curve.

Fountain JH, Lappin SL. Physiology, Renin Angiotensin System. 2022 Jun 18. In: StatPearls [Internet]. Treasure Island (FL): StatPearls Publishing; 2022 Jan–. PMID: 29261862.

Figure 8. Immunohistochemistry for Angiotensin II in kidney tissue. A. Ang II marker in convoluted tubules in WT and MFS groups. Qualitatively observed the wide stronger marker in the WT group. However, in the MFS group, the marker of the Ang II showed positive in some convoluted tubules. B. Quantification of the intensity of the Immunohistochemistry of the Ang II showed a significant reduction in the MFS group when compared to the WT group. Positive marker in brown. (***) p <0.001. The statistical analysis was performed by the Mann-Whitney test (C, E, and G).

The last comments:

One point that is not well explained is when authors affirm that fragmentation of elastic fibers in the aorta is related to an increase in collagen type I…(Discussion section). In fact, usually (not always) collagen expression is increased to compensate in some extent the loss of elasticity

As recommended, we changed the affirmation to a suggestion concerning collagen deposition in kidney tissue.

Reviewer #2: This is an interesting study providing compelling evidence of alterations in kidney morphology in a mutant fibrillin-1 mouse model. The authors clearly show that glomerular morphology is affected, and that fibrillin and elastin fibers are damaged in the Fbn1 mutants, associated with an increased deposit of collagen fibers. Renal blood flow was significantly reduced in the Fbn1 mutants, associated with an increase in microvascular density. Nevertheless, kidney function was apparently not affected in the mutant mouse model.

While the morphological phenotype in the kidney of the Fbn1 mutant mice is clearly demonstrated, it is surprising that no functional differences were shown. The filtration function of the glomeruli does not appear to be decreased in the Fbn1 mutant mice, since serum creatinine and urea are not different. The authors do not clearly address the possible reasons why this discrepancy exists between the striking morphological phenotype together with the strong reduction in renal blood flow and the lack of a functional phenotype in the kidney. Besides the important filtration function, the kidney also plays a central role in blood pressure regulation. Renal blood flow is known to affect renin production. Did the authors look at activity of the RAAS in the Fbn1 mutant mice? At a minimum, blood pressure measurements should be performed to investigate whether the reduction in renal blood flow leads to angiotensin-induced hypertension. This could be an important mechanism which might contribute to the cardiovascular pathophysiology in patients with Marfan syndrome (e.g. as described in DOI: 10.1007/s004670050327). The authors make a similar statement at the end of the discussion, but this is currently not supported by the data presented in the manuscript.

A minor comment relates to the choice of the mgΔlpn Fbn1 mutant model for this study. What was the rationale of the authors to use this specific model instead of more commonly used Fbn1 mutant mouse models? It would also be interesting to provide a more direct comparison to the previously published findings in the mgR/mgR model (DOI: 10.1007/s00428-004-1081-6), which seems quite similar to the phenotype described in this study.

Other minor comments:

- In the results section on “Fibers of the Elastic Fiber System (FEFS) in the glomerulus”, the authors mention the existence of “an amorphous substance between the mesangial area and glomerular capillary”. Can they elaborate on the potential type of substance observed here?

- In the cluster analysis of the FEFS, the authors state that subgroup A consists of microfibrils, and subgroup C of elastic fibers. What does subgroup B represent? And why were the MFS FEFS not divided in corresponding subgroups for the comparisons in Figure 2?

- From the histological analysis shown in panel A of Figure 6, the fragmentation of the elastic fibers is not very clear. Perhaps a more detailed view can be shown, or the elastic fibers can be marked more clearly?

- A limitation of the study is that only male mice were examined. Considering the known differences in the cardiovascular system, and especially related to the RAAS the authors should at least acknowledge this limitation.

We are really pleased with the reviewer's assessment of our article “This is an interesting study providing compelling evidence of alterations in kidney morphology in a mutant fibrillin-1 mouse model”.

The Reviewer makes comments relating to explaining some morphologic/physiologic aspects.

While the morphological phenotype in the kidney of the Fbn1 mutant mice is clearly demonstrated, it is surprising that no functional differences were shown. The filtration function of the glomeruli does not appear to be decreased in the Fbn1 mutant mice, since serum creatinine and urea are not different. The authors do not clearly address the possible reasons why this discrepancy exists between the striking morphological phenotype together with the strong reduction in renal blood flow and the lack of a functional phenotype in the kidney.

Yes, we understand and acknowledge that it is important that you question the reason for the absence of renal dysfunction in the face of important morphological alterations, even those associated with reduced blood flow to the renal territory. A logical hypothesis could be developed regarding the ratio of the impaired and remaining integrity of the preserved renal functional units to the point where renal failure would occur. This theory was reinforced by the animals' apparently healthy clinical appearance and normal creatinine and urea levels, indicating preserved glomerular filtration in the absence of fibrosis in the renal tissue by histology, which is considered to be the final common pathway in the evolution of virtually all types of chronic kidney disease (Li et al., 2022). Certainly, the general findings allow us to hypothesize that the process of deterioration is ongoing but still compensated by the biological functional adaptive capacity.

Li, L., Fu, H. & Liu, Y. The fibrogenic niche in kidney fibrosis: components and mechanisms. Nat Rev Nephrol 18, 545–557 (2022). https://doi.org/10.1038/s41581-022-00590-z. 

Renal blood flow is known to affect renin production. Did the authors look at activity of the RAAS in the Fbn1 mutant mice?

We have appreciated your suggestion to look at the activity of the RAAS in this study. In this study, we choose angiotensin II, because is considered one major compound of the RAAS (Fountain and Lappin 2022). In addition, the increase in Ang II has been associated with hypertension and the decrease in Ang II has been associated with a reduction in blood pressure, a reduction of blood flow, and a reduction in arterial tones (Prieto et al. 2021; Fountain and Lappin 2022). 

In this study, we study the angiotensin II expression in kidney tissue via immunohistochemistry, and we measure the intensity of the marker by Zen software.

The marker of Ang II in kidney tissue was predominantly identified in the convoluted tubules in WT. On the other hand, Ang II expression was significantly reduced in the MFS group as compared to the WT.

These findings were consistent with those of Fountain and Lappin (2022), who observed a reduction of Ang II by angiotensinogen-converting enzyme inhibitors and angiotensin receptors, leading to a reduction in blood pressure, a reduction of blood flow, and a reduction in the arterial tones. Based on these findings, the significant reduction of Ang II in the MFS group appears to explain the reduction of renal artery blood flow and velocity in microcirculation in mgΔlpn mice, as well as reductions of the systolic peak in the aortic spectral curve.

Fountain JH, Lappin SL. Physiology, Renin Angiotensin System. 2022 Jun 18. In: StatPearls [Internet]. Treasure Island (FL): StatPearls Publishing; 2022 Jan–. PMID: 29261862.

Prieto MC, Gonzalez AA, Visniauskas B, Navar LG. The evolving complexity of the collecting duct renin-angiotensin system in hypertension. Nat Rev Nephrol. 2021 Jul;17(7):481-492. doi: 10.1038/s41581-021-00414-6.

Figure 8. Immunohistochemistry for Angiotensin II in kidney tissue. A. Ang II marker in convoluted tubules in WT and MFS groups. Qualitatively observed the wide stronger marker in the WT group. However, in the MFS group, the marker of the Ang II showed positive in some convoluted tubules. B. Quantification of the intensity of the Immunohistochemistry of the Ang II showed a significant reduction in the MFS group when compared to the WT group. Positive marker in brown. (***) p <0.001. The statistical analysis was performed by the Mann-Whitney test (C, E, and G).

At a minimum, blood pressure measurements should be performed to investigate whether the reduction in renal blood flow leads to angiotensin-induced hypertension. This could be an important mechanism that might contribute to the cardiovascular pathophysiology in patients with Marfan syndrome (e.g. as described in DOI: 10.1007/s004670050327). The authors make a similar statement at the end of the discussion, but this is currently not supported by the data presented in the manuscript.

Despite the strong correlation between renal injury and blood pressure, mean arterial pressure was not evaluated in this study due to the possibility that invasive probe placement (Drost CJ 2019) could negatively affect renal micro- and macrocirculatory hemodynamics. However, the blood flow and spectral curve of the abdominal aorta were evaluated by placing a periaortic probe around it right above the inferior infra-phrenic artery. 

Data analysis showed a significant reduction in aortic flow in the MFS group compared to the wild-type group (Figure below - B), thus justifying the reduction in renal artery flow in the MFS group. We have previously reported the aorta blood flow reduction in mg∆lpn female mice (de Souza et al 2019) and it was attributed to high-output cardiac failure.

Additionally, we noticed continuous peak systolic velocity in the WT group as well as constant monophasic flow patterns and no alterations in the patterns of the relaxation curves. As opposed to the WT group, the MFS group displayed monophasic flow with variable peak systolic velocities and a lower systolic peak. Moreover, it was followed by altered diastolic phase peaks (Figure below - A).

Drost CJ. This handbook is an educational service provided. Transonic Systems Inc.®.2019.

Supplementary figure 2. Aorta Blood Flow. A. Spectral curve analysis of aortic blood flow, in the WT group, presents a uniform monophasic flow pattern with similar peak systolic velocities (PSVs) (black arrows) and no alterations in relaxation curves (black arrowhead). In MFS- we observe a PSV reduced (blue arrow), and alteration in the diastole curve (blue arrowhead). B. Quantification of the aorta blood flow revealed a significantly decreased in the MFS group when compared to the WT group. (*** ρ<0.0001) The statistical analysis was performed by the Mann-Whitney test

A minor comment relates to the choice of the mgΔlpn Fbn1 mutant model for this study. What was the rationale of the authors to use this specific model instead of more commonly used Fbn1 mutant mouse models? It would also be interesting to provide a more direct comparison to the previously published findings in the mgR/mgR model (DOI: 10.1007/s00428-004-1081-6), which seems quite similar to the phenotype described in this study

We chose the mgΔlpn mice model because it has been generated by our group (Lima et al. 2010), and it is a dominant-negative model where heterozygotes present classical MFS phenotypes, including skeletal, ocular, and cardiovascular manifestations. In particular, the mgΔlpn presents aneurysm and aortic dissection similar to MFS patients (Lima et al. 2010; de Souza et al 2019; Souza et al., 2021a; Souza et a., 2021b). In contrast, the mgR/mgR mouse model has a hypomorphic FBN1 mutation, and only homozygotes present skeletal and cardiovascular phenotypes. 

We appreciate your suggestion to compare direct from both models mgR/mg R and mgΔlpn. We added in the discussion sections the comparison of models. 

Line 303 to 308 “Although in the MFS group we observed alterations in structural components, no changes in renal functions were discovered. Hartner et al. [10] found in the mgR/mgR model, with lower expression of fibrillin-1, glomerular hypoplasia with decreased immunolocalization of fibrillin-1 in the renal parenchyma, as well as intact renal functionality. These findings support the hypothesis that alterations in fibrillin-1 are insufficient to cause a change in renal function, but they do cause an alteration in kidney structure”.

Franken, R., Heesterbeek, T. J., de Waard, V., Zwinderman, A. H., Pals, G., Mulder, B. J., & Groenink, M. (2014). Diagnosis and genetics of Marfan syndrome. Expert opinion on orphan drugs, 2(10), 1049-1062.

- In the results section on “Fibers of the Elastic Fiber System (FEFS) in the glomerulus”, the authors mention the existence of “an amorphous substance between the mesangial area and glomerular capillary”. Can they elaborate on the potential type of substance observed here?

The amorphous substances, which we indicated in TEM analysis could be possibly a basement membrane-like material (BM), due to localization. The presence of the amorphous material was referenced in subendothelial alterations in the glomerular basement membrane (Sbar et al., 1996 e Hsu e Churg 1979). We added in the manuscript the suggestion. 

Line 166 to 168: … “Possibly the amorphous substance inside the glomerulus near the glomerular capillaries is a basement membrane-like material (BM). This sort of substance has already been identified in glomerular basement membrane subendothelial alterations. (Hsu e Churg 1979; Sbar et al., 1996)”

- In the cluster analysis of the FEFS, the authors state that subgroup A consists of microfibrils, and subgroup C of elastic fibers. What does subgroup B represent? And why were the MFS FEFS not divided in corresponding subgroups for the comparisons in Figure 2?

The cluster analysis automatically separated the elastic fiber measurements into three subgroups, of which, coincidentally, subgroup A has a similar thickness distribution to microfibrils and subgroup C to mature elastic fibers. Possibly, subgroup B can be considered a group of elastic fibers that are in the process of elastogenesis, or even elauninin fibers. However, morphologically we did not identify the classic aspects of elauninin fibers. 

The cluster analysis was not performed in the MFS group. Because the disease can lead to a dubious structural feature of the elastic fibers without a precise understanding of the probable modifications. Possibly the changes in fibrillin-1, which in itself already alters the formation of elastic fibers, and possibly the proteolytic actions, such as the increased expression of MMP-9 (an enzyme that can degrade elastic fibers and interfere with elastogenesis).Both scenarios might lead to a dubious structural feature of the elastic fibers without a precise understanding of the probable modifications.

By automatically separating the WT group into three subgroups, we observed that there are different FEFS sizes, possibly indicating a complexity in the structural arrangements of the elastic fibers in the kidney. On the other hand, when comparing the three subgroups with the MFS group, we observed that the FEFS of the MFS group has a significant reduction when compared to subgroup C, indicating that the elastic fibers of the MFS group do not have the capacity to form mature elastic fibers. In addition, when we apply cluster analysis, we observed in the MFS group the subgroup predominance is B.

- From the histological analysis shown in panel A of Figure 6, the fragmentation of the elastic fibers is not very clear. Perhaps a more detailed view can be shown, or the elastic fibers can be marked more clearly?

We appreciate the suggestion to change the microphotographs to contribute to the work. Attached we have added this photo to the image for a better representation of changes in the elastic fibers in the renal artery.

---

## [Decision Letter · Decision Letter 1]

30 Jan 2023

PONE-D-22-28441R1Extracellular Matrix and Vascular Dynamics in the Kidney of a murine model for Marfan syndromePLOS ONE

Dear Dr. Pereira,

Thank you for submitting your manuscript to PLOS ONE. After careful consideration, we feel that it has merit but does not fully meet PLOS ONE’s publication criteria as it currently stands. Therefore, we invite you to submit a revised version of the manuscript that addresses the points raised during the review process.

We look forward to receiving your revised manuscript.

Kind regards,

Peter R. Corridon

Academic Editor

PLOS ONE

Journal Requirements:

Additional Editor Comments:

During the review, points were made related to the conclusions drawn from the kidney AngII staining. It is recommended that the authors perform plasma renin concentration or plasma renin activity measurements to address this remaining issue conclusively. 

Reviewers' comments:

Reviewer's Responses to Questions

**Comments to the Author**

1. If the authors have adequately addressed your comments raised in a previous round of review and you feel that this manuscript is now acceptable for publication, you may indicate that here to bypass the “Comments to the Author” section, enter your conflict of interest statement in the “Confidential to Editor” section, and submit your "Accept" recommendation.

Reviewer #1: All comments have been addressed

Reviewer #2: (No Response)

2. Is the manuscript technically sound, and do the data support the conclusions?

Reviewer #1: Yes

Reviewer #2: Partly

3. Has the statistical analysis been performed appropriately and rigorously? 

Reviewer #1: I Don't Know

Reviewer #2: Yes

4. Have the authors made all data underlying the findings in their manuscript fully available?

Reviewer #1: Yes

Reviewer #2: Yes

5. Is the manuscript presented in an intelligible fashion and written in standard English?

Reviewer #1: Yes

Reviewer #2: Yes

6. Review Comments to the Author

Reviewer #1: All my comments have been satisfactorily addressed both in the response letter and the resubmitted manuscript

Reviewer #2: In the revised version the authors have provided some more experimental data to support their conclusions. The additional blood flow measurements and immunohistochemical staining of AngII are appreciated, but unfortunately do not entirely address the important question regarding the potential systemic effects of reduced renal blood flow.

On one hand, the measured reduction in aortic blood flow does indeed argue against a hypertensive phenotype in the mgΔlpn model. On the other hand, blood pressure is also dependent on vascular tone and it is possible that chronic hypertension has led to remodeling of the vascular wall, increasing stiffness and reducing luminal diameter, resulting in decreased flow despite elevated or maintained blood pressure. It is appreciated that invasive measurements of blood flow are not advised in this model, but perhaps non-invasive (tail-cuff) measurements are plausible in future studies, although they admittedly do lack the same level of precision as invasive measurements. It is recommended that the authors do not make definitive statements about systemic blood pressure without these measurements.

Another final concern is that the local kidney AngII levels which are suggested by the IHC stain shown in the revised manuscript do not necessarily reflect systemic RAAS activity. The intrarenal production of AngII can be considered as a separate system, independent of renin release from juxtaglomerular cells. Some evidence suggests that intrarenal AngII is predominantly regulated by angiotensinogen availability, since kidney renin activity is approximately 1000-fold higher than plasma renin activity. It is also not correct to link changes in kidney AngII to effects on aortic and renal blood flow, which are regulated by plasma AngII levels. Therefore, I recommend that the authors measure plasma renin concentration or plasma renin activity in this mouse model, particularly since reduced renal blood flow is the classical trigger for renin release from juxtaglomerular cells, which could then contribute to vessel damage by increased systemic AngII production.

7. PLOS authors have the option to publish the peer review history of their article (what does this mean?). If published, this will include your full peer review and any attached files.

Reviewer #1: **Yes: **Gustavo Egea

Reviewer #2: No

---

## [Author Response · Author response to Decision Letter 1]

12 Mar 2023

Review Comments to the Author

We appreciate the reviewers' efforts in providing a new opportunity to improve the text and by providing constructive ideas. To accommodate all of their recommendations, we made significant adjustments to the text, which we address below. We hope that our paper is now of acceptable quality to be published as a PlosOne article.

Reviewer #1: All my comments have been satisfactorily addressed both in the response letter and the resubmitted. 

We are very grateful for the pertinent comments that brought improvements to our work.

Reviewer #2: In the revised version the authors have provided some more experimental data to support their conclusions. The additional blood flow measurements and immunohistochemical staining of AngII are appreciated, but unfortunately do not entirely address the important question regarding the potential systemic effects of reduced renal blood flow.

On one hand, the measured reduction in aortic blood flow does indeed argue against a hypertensive phenotype in the mgΔlpn model. On the other hand, blood pressure is also dependent on vascular tone and it is possible that chronic hypertension has led to remodeling of the vascular wall, increasing stiffness and reducing luminal diameter, resulting in decreased flow despite elevated or maintained blood pressure. It is appreciated that invasive measurements of blood flow are not advised in this model, but perhaps non-invasive (tail-cuff) measurements are plausible in future studies, although they admittedly do lack the same level of precision as invasive measurements. It is recommended that the authors do not make definitive statements about systemic blood pressure without these measurements.

Another final concern is that the local kidney AngII levels which are suggested by the IHC stain shown in the revised manuscript do not necessarily reflect systemic RAAS activity. The intrarenal production of AngII can be considered as a separate system, independent of renin release from juxtaglomerular cells. Some evidence suggests that intrarenal AngII is predominantly regulated by angiotensinogen availability, since kidney renin activity is approximately 1000-fold higher than plasma renin activity. It is also not correct to link changes in kidney AngII to effects on aortic and renal blood flow, which are regulated by plasma AngII levels. Therefore, I recommend that the authors measure plasma renin concentration or plasma renin activity in this mouse model, particularly since reduced renal blood flow is the classical trigger for renin release from juxtaglomerular cells, which could then contribute to vessel damage by increased systemic AngII production.

We appreciate the reviewer's comments about adding the new data "In the revised version, the authors have provided some more experimental data to support their conclusions. The additional blood flow measurements and immunohistochemical staining of AngII are appreciated"

On one hand, the measured reduction in aortic blood flow does indeed argue against a hypertensive phenotype in the mgΔlpn model. On the other hand, blood pressure is also dependent on vascular tone and it is possible that chronic hypertension has led to remodeling of the vascular wall, increasing stiffness and reducing luminal diameter, resulting in decreased flow despite elevated or maintained blood pressure. It is appreciated that invasive measurements of blood flow are not advised in this model, but perhaps non-invasive (tail-cuff) measurements are plausible in future studies, although they admittedly do lack the same level of precision as invasive measurements. It is recommended that the authors do not make definitive statements about systemic blood pressure without these measurements.

We appreciate your interpretation of the relationship between the changed vascular tone and the reduced blood flow. 

We were grateful for the proposal to assess systemic blood pressure using non-invasive (tail-cuff) readings; this is something we will do in future studies.

We'll revise the content to refrain from making categorical claims regarding systemic blood pressure.

Another final concern is that the local kidney AngII levels which are suggested by the IHC stain shown in the revised manuscript do not necessarily reflect systemic RAAS activity. The intrarenal production of AngII can be considered as a separate system, independent of renin release from juxtaglomerular cells. Some evidence suggests that intrarenal AngII is predominantly regulated by angiotensinogen availability, since kidney renin activity is approximately 1000-fold higher than plasma renin activity. It is also not correct to link changes in kidney AngII to effects on aortic and renal blood flow, which are regulated by plasma AngII levels. Therefore, I recommend that the authors measure plasma renin concentration or plasma renin activity in this mouse model, particularly since reduced renal blood flow is the classical trigger for renin release from juxtaglomerular cells, which could then contribute to vessel damage by increased systemic AngII production.

We appreciate your comments and suggestions about Ang II and renin plasma activity, respectively. However, the animals' plasma was not stored, making it impossible to measure any plasma components, as mentioned in the last review. Additionally, 6-month-old animals will be needed in order to analyze plasma renin, necessitating significantly longer study duration.

For this reason, we changed the text of the manuscript and made no statement about the relationship between systemic blood pressure and the activity of RAAS components. Besides, we would like to mention that this study was mainly focused on the descriptive aspect of the kidney in the mgΔlpn model, which may awaken as a finding for future studies, in addition to enabling new therapeutic targets and providing an insight into the pathophysiological mechanism of Marfan syndrome disease.

Line 362-370: “Furthermore, in the MFS group, a significant reduction of angiotensin II (AngII) was found in the renal convoluted tubule, which at first raised the possibility of a direct connection with the blood flow and arterial tone alterations. Alterations in intrarenal AngII have been described in some physiological conditions, such as in renal disfunction and hypertension. However, the precise contribution of intrarenal AngII in these conditions is poorly understood [63, 64, 65, 66]. As we did not measure the plasmatic components of the Renin-Angiotensin-Aldosterone system (RAAS) and blood pressure, we could not conjecture their relationship with the hemodynamic state. In future studies, these pathophysiological aspects need to be better exploited.”

63. Zhuo JL, Li XC. Novel roles of intracrine angiotensin II and signalling mechanisms in kidney cells. J Renin Angiotensin Aldosterone Syst. 2007 Mar;8(1):23-33. doi: 10.3317/jraas.2007.003.

64. Zhuo JL, Ferrao FM, Zheng Y, Li XC. New frontiers in the intrarenal Renin-Angiotensin system: a critical review of classical and new paradigms. Front Endocrinol (Lausanne). 2013 Nov 11;4:166. doi: 10.3389/fendo.2013.00166.

65. Li XC, Zhu D, Zheng X, Zhang J, Zhuo JL. Intratubular and intracellular renin-angiotensin system in the kidney: a unifying perspective in blood pressure control. Clin Sci (Lond). 2018 Jul 9;132(13):1383-1401. doi: 10.1042/CS20180121.

66. Navar LG, Prieto MC, Satou R, Kobori H. Intrarenal angiotensin II and its contribution to the genesis of chronic hypertension. Curr Opin Pharmacol. 2011 Apr;11(2):180-6. doi: 10.1016/j.coph.2011.01.009.

---

## [Decision Letter · Decision Letter 2]

23 Mar 2023

PONE-D-22-28441R2Extracellular Matrix and Vascular Dynamics in the Kidney of a murine model for Marfan syndromePLOS ONE

Dear Dr. Pereira,

Thank you for submitting your manuscript to PLOS ONE. After careful consideration, we feel that it has merit but does not fully meet PLOS ONE’s publication criteria as it currently stands. Therefore, we invite you to submit a revised version of the manuscript that addresses the points raised during the review process.

Authors, please upload the correct supplementary figures. I also suggest that the manuscript is still carefully checked by a native English speaker to make some more grammatical improvements.

We look forward to receiving your revised manuscript.

Kind regards,

Peter R. Corridon

Academic Editor

PLOS ONE

Journal Requirements:

Reviewers' comments:

Reviewer's Responses to Questions

**Comments to the Author**

1. If the authors have adequately addressed your comments raised in a previous round of review and you feel that this manuscript is now acceptable for publication, you may indicate that here to bypass the “Comments to the Author” section, enter your conflict of interest statement in the “Confidential to Editor” section, and submit your "Accept" recommendation.

Reviewer #2: (No Response)

2. Is the manuscript technically sound, and do the data support the conclusions?

Reviewer #2: Yes

3. Has the statistical analysis been performed appropriately and rigorously? 

Reviewer #2: Yes

4. Have the authors made all data underlying the findings in their manuscript fully available?

Reviewer #2: Yes

5. Is the manuscript presented in an intelligible fashion and written in standard English?

Reviewer #2: No

6. Review Comments to the Author

Reviewer #2: I agree with the proposed changes to the discussion, weakening the statements regarding the connection between the intrarenal Ang II measurements and the hemodynamic state in the mice. I would suggest to also rephrase the rationale given on lines 289-292, where the link between Ang II and vascular tone / arterial pressure is highlighted. As the authors acknowledge in the revised discussion, intrarenal Ang II measurements will not address this issue, so this statement should also be amended not to confuse the reader.

I have noticed that Supplementary Figure 2 (showing the aortic flow profile) is missing from the submission. There is a reference to it in the text and a legend is provided, but currently Fig S2 shows the creatinine and urea measurements, and Fig S3 shows the SDF video processing.

Finally, It would also be good if the manuscript can be checked one last time by a native English speaker, as some sentences still contain some grammatical mistakes, particularly in the revised parts of the text.

7. PLOS authors have the option to publish the peer review history of their article (what does this mean?). If published, this will include your full peer review and any attached files.

Reviewer #2: No

---

## [Author Response · Author response to Decision Letter 2]

11 Apr 2023

Review Comments to the Author

We appreciate the reviewers' efforts in providing a new opportunity to improve the text. To accommodate all of their recommendations, we made significant adjustments to the text and had it reviewed by a native English speaker. We hope that our paper is now of acceptable quality to be published as a PlosOne article.

Reviewer #2: I agree with the proposed changes to the discussion, weakening the statements regarding the connection between the intrarenal Ang II measurements and the hemodynamic state in the mice. I would suggest to also rephrase the rationale given on lines 289-292, where the link between Ang II and vascular tone / arterial pressure is highlighted. As the authors acknowledge in the revised discussion, intrarenal Ang II measurements will not address this issue, so this statement should also be amended not to confuse the reader. 

We are grateful for the suggestions to improve the manuscript. We changed the text in lines 289 to 291. 

Lines 289-291: Furthermore, due to significant reductions in blood flow in the MFS group, we analyzed angiotensin II (Ang II) in the kidney. Ang II is a major component of the renin-angiotensin-aldosterone system (RAAS), produced by convoluted tubules and juxtaglomerular cells [43, 44].

I have noticed that Supplementary Figure 2 (showing the aortic flow profile) is missing from the submission. There is a reference to it in the text and a legend is provided, but currently Fig S2 shows the creatinine and urea measurements, and Fig S3 shows the SDF video processing. 

We apologize for the mistake; we included the necessary supplementary material.

Finally, It would also be good if the manuscript can be checked one last time by a native English speaker, as some sentences still contain some grammatical mistakes, particularly in the revised parts of the text.

We appreciate the reviewers’ comments. The Text was reviewed by a native English speaker.

---

## [Editor Report · Decision Letter 3]

24 Apr 2023

Extracellular Matrix and Vascular Dynamics in the Kidney of a murine model for Marfan syndrome

PONE-D-22-28441R3

Dear Dr. Pereira,

We’re pleased to inform you that your manuscript has been judged scientifically suitable for publication and will be formally accepted for publication once it meets all outstanding technical requirements.

Kind regards,

Peter R. Corridon

Academic Editor

PLOS ONE
---

## [Editor Report · Acceptance letter]

28 Apr 2023

PONE-D-22-28441R3 

Extracellular Matrix and Vascular Dynamics in the Kidney of a murine model for Marfan syndrome 

Dear Dr. Pereira:

I'm pleased to inform you that your manuscript has been deemed suitable for publication in PLOS ONE. Congratulations! Your manuscript is now with our production department. 

Kind regards, 

on behalf of

Dr. Peter R. Corridon 

Academic Editor

PLOS ONE